# A UNIFIED FRAMEWORK FOR COMPARING LEARNING ALGORITHMS

## ABSTRACT

We propose a framework for *(learning) algorithm comparisons*, wherein the goal is to find similarities and differences between models trained with two different learning algorithms. We begin by formalizing the goal of algorithm comparison as finding *distinguishing feature transformations*, input transformations that change the predictions of models trained with one learning algorithm but not the other. We then present a two-stage method for algorithm comparisons based on comparing how models use the training data, leveraging the recently proposed datamodel representations (Ilyas et al., 2022). We demonstrate our framework through three case studies that compare models trained with/without standard data augmentation, with/without pre-training, and with different optimizer hyperparameters.

## 1 INTRODUCTION

Building a machine learning model involves a series of design choices. Indeed, even after choosing a dataset, for example, one must decide on a model architecture, an optimization method, and a data augmentation pipeline. These design choices together define a *learning algorithm*, a function mapping training datasets to machine learning models.

Even if they do not affect accuracy, design choices determine the *biases* of the resulting models. For example, Hermann et al. (2020) find significant variation in shape bias (Geirhos et al., 2019) across a group of ImageNet models that vary in accuracy by less than 1%. In order to understand the impact of design choices, we thus need to be able to differentiate learning algorithms in a more fine-grained way than accuracy alone.

Motivated by this observation, we develop a unified framework for comparing learning algorithms. Our proposed framework comprises (a) a precise, quantitative definition of learning algorithm comparison; and (b) a concrete methodology for comparing any two algorithms. For (a), we frame the algorithm comparison problem as one of finding input transformations that *distinguish* the two algorithms. This goal is different and more general than quantifying model similarity (Ding et al., 2021; Bansal et al., 2021; Morcos et al., 2018a) or testing specific biases (Hermann et al., 2020). For (b), we propose a two-stage method for comparing algorithms in terms of *how they use the training data*.

In the first stage of this method, we leverage *datamodel representations* (Ilyas et al., 2022) to find weighted combinations of training examples (which we call *training directions*) that have disparate impact on test-time behavior of models across learning algorithms. In the second stage, we filter the subpopulation of test examples that are most influenced by each identified training direction, then manually inspect them to infer a shared feature (e.g., we might notice that all of the test images contain a spider web). We then tie this intuition back to our quantitative definition by designing a distinguishing feature transformation based on the shared feature (e.g., overlaying a spider web pattern to the background of an image).

We illustrate the utility of our framework through three case studies (Section 3), motivated by typical choices one needs to make within machine learning pipelines, namely:

- **Data augmentation:** We compare classifiers trained with and without data augmentation on the LIVING17 (Santurkar et al., 2021) dataset. We show that models trained *with* data augmentation, while having higher overall accuracy, are *more* prone to picking up specific instances of co-occurrence bias and texture bias compared to models trained *without* data augmentation. For

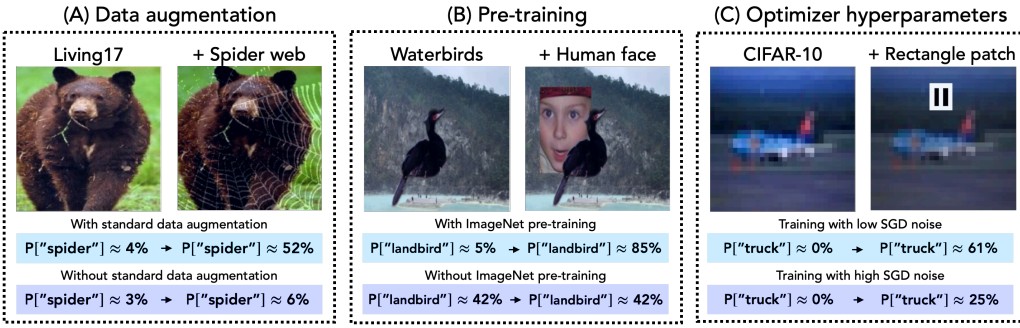

Figure 1: A visual summary of our case studies. We use our method to study the differences between training with and without standard data augmentation, with and without ImageNet pre-training, and with different choices of SGD optimizer hyperparameters. In all three cases, our framework allows us to pinpoint concrete ways in which the two algorithms being compared differ.

instance, adding a spider web (Figure 1A)—which co-occurs with spiders—to random images increases the "spider" confidence of augmentation-trained models by 15% on average, but increases the confidence of non-augmented models by less than 1%.

- **ImageNet pre-training:** We compare classifiers first pre-trained on ImageNet (Deng et al., 2009; Russakovsky et al., 2015) then fine-tuned on WATERBIRDS (Sagawa et al., 2020) with classifiers trained from scratch on WATERBIRDS. We demonstrate that pre-training can either suppress or amplify specific spurious correlations. As an example of the former, adding a yellow patch to random images increases confidence in the "landbird" label of models trained from scratch by 12%, but actually decreases the confidence of pre-trained models by 4%. As an example of the latter, adding a human face (Figure 1B) to the background increases the "landbird" confidence of pre-trained models by 4%, but decreases the confidence of models trained from scratch by 1%.

- **Optimizer hyperparameters:** Finally, we compare classifiers trained on CIFAR-10 (Krizhevsky, 2009) using stochastic gradient descent (SGD) with different choices of learning rates and batch sizes. Our analysis pinpoints subtle differences in model behavior induced by small changes to these hyperparameters. For example, adding a small pattern that resembles windows (Figure 1C) to random images increases the "truck" confidence by 7% on average for models trained with a smaller learning rate, but increases the confidence by only 2% for models trained with a larger learning rate.

Across all three case studies, our framework surfaces fine-grained differences between models trained with different learning algorithms, enabling us to better understand the role of the design choices that make up a learning algorithm.

## 2 COMPARING LEARNING ALGORITHMS

In this section, we describe our (learning) algorithm comparison framework. In Section 2.1, we formalize algorithm comparison as the task of identifying *distinguishing transformations*. These are functions that—when applied to test examples—significantly and consistently change the predictions of one model class but not the other. In Section 2.2, we describe our method for identifying distinguishing feature transformations by comparing how each model class uses the training data.

### 2.1 FORMALIZING ALGORITHM COMPARISONS VIA DISTINGUISHING TRANSFORMATIONS

The goal of algorithm comparison is to understand the ways in which two learning algorithms (trained on the same data distribution) differ in the models they yield. More specifically, we are interested in comparing the *model classes* induced by the two learning algorithms:

**Definition 1** (Induced model class). *Given input space $\mathcal{X}$, label space $\mathcal{Y}$, and model space $\mathcal{M} \subset \mathcal{X} \to \mathcal{Y}$, a learning algorithm $\mathcal{A} : (\mathcal{X} \times \mathcal{Y})^* \to \mathcal{M}$ is a (potentially random) function mapping a set of input-label pairs to a model. Fixing a data distribution $\mathcal{D}$, the model class induced by algorithm $\mathcal{A}$ is the distribution over $\mathcal{M}$ that results from applying $\mathcal{A}$ to randomly sampled datasets from $\mathcal{D}$.*

The perspective we adopt here is that model classes differ insofar as they use different features to make predictions. We make this notion precise by defining functions that we call *distinguishing transformations*:

**Definition 2** (Distinguishing feature transformation). *Let $\mathcal{A}_1, \mathcal{A}_2$ denote learning algorithms, $S$ a dataset of input-label pairs, and $\mathcal{L}$ a loss function (e.g., correct-class margin). Suppose $M_1$ and $M_2$ are models trained on dataset $\mathcal{D}$ using algorithms $\mathcal{A}_1$ and $\mathcal{A}_2$ respectively. Then, a $(\epsilon, \delta)$-distinguishing feature transformation of $M_1$ with respect to $M_2$ is a function $F : \mathcal{X} \to \mathcal{X}$ such that for some label $y_c \in \mathcal{Y}$,*

$$\overbrace{\mathbb{E}[L_1(F(x), y_c) - L_1(x, y_c)] \geq \delta}^{\text{Counterfactual effect of } F \text{ on } M_1} \qquad and \qquad \overbrace{\mathbb{E}[L_2(F(x), y_c) - L_2(x, y_c)] \leq \epsilon,}^{\text{Counterfactual effect of } F \text{ on } M_2}$$

*where $L_i(x, y) = \mathcal{L}(M_i(x), y)$, and the expectations above are taken over both inputs $x$ and randomness in the learning algorithm.*

Intuitively, a distinguishing feature transformation is just a function $F$ that, when applied to test data points, significantly changes the predictions of one model class—but not the other—in a consistent way. Definition 2 also immediately suggests a way to *evaluate* the effectiveness of a distinguishing feature transformation. That is, given a hypothesis about how two algorithms differ (e.g., that models trained with $\mathcal{A}_1$ are more sensitive to texture than those trained with $\mathcal{A}_2$), one can design a corresponding transformation $F$ (e.g., applying style transfer, as in Geirhos et al. (2019)), and directly measure its relative effect on the two model classes.

***Informative* distinguishing feature transformations.** Not every distinguishing transformation $F$ sheds the same amount of light on model behavior. For example, given any two non-identical learning algorithms, we could craft a transformation $F$ that imperceptibly modifies its input to satisfy Definition 2 (e.g., by using adversarial examples). Alternatively, one could craft an $F$ that arbitrarily transforms its inputs into pathological out-of-distribution examples on which the two model classes disagree. While these transformations satisfy Definition 2, they yield no benefit in terms of qualitatively understanding the differences in *salient* features used by the model classes. More concretely, an *informative* distinguishing feature transformation must (a) capture a feature that naturally arises in the data distribution and (b) be semantically meaningful.

## 2.2 IDENTIFYING DISTINGUISHING FEATURE TRANSFORMATIONS

We now describe our two-stage method for comparing learning algorithms based on *how they use the training data*. Recall that our goal is to identify *informative* distinguishing transformations—that is, those that (a) capture a feature that arises naturally in the data distribution and (b) are semantically meaningful. Rather than search for distinguishing feature transformations directly, we instead first identify *training directions*, or weighted combinations of training examples, that impact test performance of models trained with one learning algorithm but not the other. Then, we use a human-in-the-loop approach to extract semantically meaningful features and corresponding transformations from these directions.

**Stage I: An algorithm for finding distinguishing training directions.** First, we find weighted combinations of training examples that influence the predictions of models trained with one learning algorithm but not the other. Our algorithm leverages *datamodel representations* (Ilyas et al., 2022) and comprises the following three steps:

1. **Compute datamodels for each algorithm.** Our point of start is computing a datamodel representation for each example in the test set $T$. Given training set $S$ and learning algorithm $\mathcal{A}$, a datamodel representation for test example $x_i$ is a vector $\theta_i \in \mathbb{R}^{|S|}$, where $\theta_{ij}$ measures the extent to which models trained with $\mathcal{A}$ depend[1] on the $j$-th training example to correctly classify example $x_i$. In other words, $\theta_i$ is a *training direction* that strongly influences the prediction for example

---

[1] For readers familiar with influence functions (Koh & Liang, 2017; Hampel et al., 2011), an intuitive (but not quite accurate) way to interpret datamodel weight $\theta_{ij}$ is as the influence of the $j$-th training example on test example $x_i$. In Appendix A, we discuss additional properties of datamodel representations that make them particularly well-suited for model comparison.

$x_i$. We compute two sets of datamodels—$\theta^{(1)}$ and $\theta^{(2)}$—corresponding to model classes induced by learning algorithms $\mathcal{A}_1$ and $\mathcal{A}_2$ respectively.

2. **Compute residual datamodels.** Next, we compute a *residual datamodel* for each test example $x_i$, which is the projection of the datamodel representation $\theta_i^{(1)}$ onto the null space of datamodel representation $\theta_i^{(2)}$:

$$\theta_i^{(1\backslash 2)} = \hat{\theta}_i^{(1)} - \langle \hat{\theta}_i^{(1)}, \hat{\theta}_i^{(2)} \rangle \, \hat{\theta}_i^{(2)},$$

where $\hat{\theta}_i = \theta_i / \|\theta_i\|_2$ denotes the normalized version of datamodel $\theta_i$. Intuitively, the residual datamodels of algorithm $\mathcal{A}_1$ with respect to $\mathcal{A}_2$ correspond to the training directions that influence $\mathcal{A}_1$ after "projecting away" the component that also influences $\mathcal{A}_2$.

3. **Run principal component analysis.** Finally, we use principal component analysis (PCA) to find the highest-variance directions in the space of residual datamodels. That is, we run

$$\ell\text{-PCA}(\{\theta_1^{(1\backslash 2)}, \ldots, \theta_{|T|}^{(1\backslash 2)}\})$$

to find the top $\ell$ principal components of the residual datamodels. Intuitively (deferring formal analysis to Appendix A), we expect the returned principal components to be the most distinguishing training directions across the test set.

We illustrate our algorithm visually in the top half of Figure 2.

**Stage II: Human-in-the-loop analysis.** With these distinguishing directions in hand, we use a human-in-the-loop analysis to identify *informative* distinguishing features transformations in three steps. First, given a principal component, we inspect the test examples whose residual datamodels are most aligned with that component. We view these examples as representing the *subpopulation* that depends most heavily on the training direction. We then use visual inspection (see, e.g., Figure 2C) and if needed, additional analysis[2], to infer a *distinguishing feature* shared by examples in the surfaced subpopulation. Finally, we design a distinguishing feature transformation (as in Figure 2D) to counterfactually verify the effect of the inferred feature on model behavior.

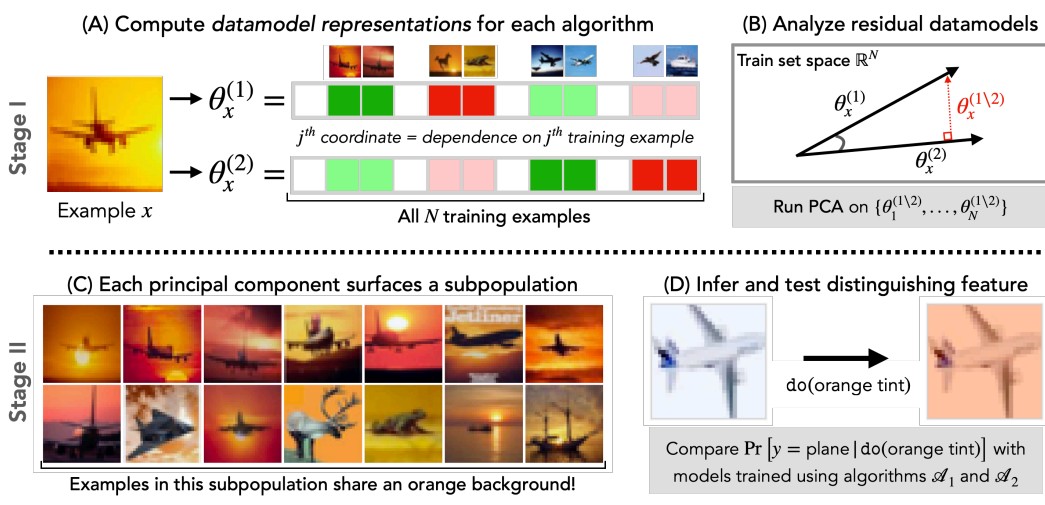

Figure 2: A visual summary of our two-stage approach to algorithm comparison. In the first stage (top row) we use examples' datamodel representations (Ilyas et al., 2022) to find so-called *distinguishing training directions*—weighted combinations of training examples that impact the two algorithms disparately across the test set. In the second stage (bottom row) we surface *subpopulations* that rely on the identified directions, and use a human-in-the-loop to go from the identified distinguishing training direction to a testable feature transformation.

---

[2]Visual inspection may be insufficient to identify a *single* distinguishing feature. In such cases, we resort to additional human-in-the-loop analysis (see Appendix D) to identify the distinguishing feature.

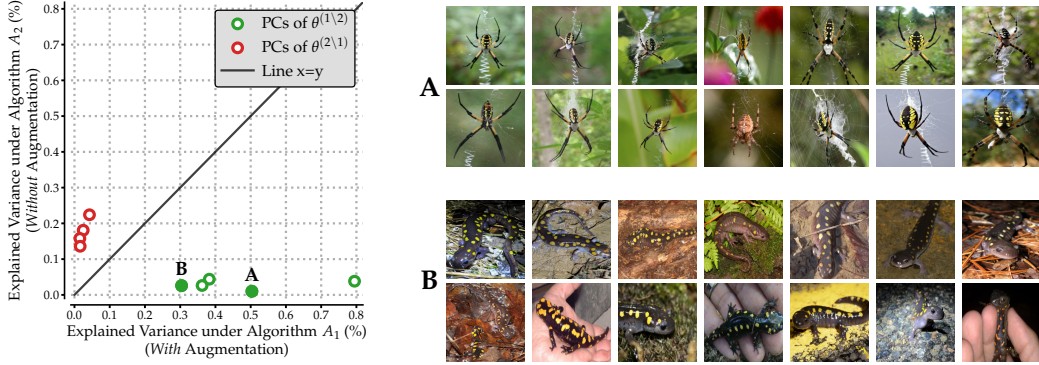

Figure 3: **Comparing LIVING17 models trained with and without data augmentation. (Left)** Each green (resp., red) point is a *training direction* (i.e., a vector $v \in \mathbb{R}^{|S|}$ representing a weighted combination of training examples) that distinguishes $\mathcal{A}_1$ from $\mathcal{A}_2$ (resp., $\mathcal{A}_2$ from $\mathcal{A}_1$) as identified by the first stage of our framework. The $x$ and $y$ coordinates of each point represent the importance of the training direction to models trained with $\mathcal{A}_1$ and $\mathcal{A}_2$ respectively. **(Right)** Test examples most impacted by the distinguishing training directions annotated **A** and **B**. **A** seems to correspond to spiders on spider webs, and **B** to salamanders with yellow polka dots.

## 3 APPLYING THE ALGORITHM COMPARISON FRAMEWORK

We demonstrate our comparison framework through a variety of case studies. Specifically, we consider three key aspects of the standard training pipeline—data augmentation, pre-training, and optimization—and characterize their effect in terms of the features that they amplify or suppress. In each case study, we follow the same procedure: (i) use the algorithm from Stage I above (Section 2) to identify *examples training directions*; (ii) inspect the test images that depend most heavily on these training directions to come up with a *candidate distinguishing transformation*; and (iii) apply these candidate transformation to test examples and compare its effect on models trained with the two learning algorithms.

### 3.1 CASE STUDY: DATA AUGMENTATION

Data augmentation is a key component of the standard computer vision training pipeline, as its application can significantly improve model performance. However, the effect of using data augmentation on models' learned *features* remains elusive. In this case study, we study the effect of data augmentation in the context of classifiers trained on the ImageNet-derived LIVING17 dataset (Santurkar et al., 2021). Specifically, we compare two classes of ResNet-18 models trained on this dataset with the exact same settings modulo the use of data augmentation. That is, we consider the following two learning algorithms:

- **Algorithm $\mathcal{A}_1$:** Training with standard augmentation, i.e., horizontal flip and resized random crop with `torchvision` default parameters. Resulting models attain 89% average test accuracy (where the average is taken over randomness in training).

- **Algorithm $\mathcal{A}_2$:** Training without data augmentation. Models attain 81% average accuracy.

Appendix C.1 provides further experimental details.

**Identifying distinguishing features.** We compare algorithms $\mathcal{A}_1$ and $\mathcal{A}_2$ using our method from Section 2. The first stage of this method finds a set of *distinguishing training directions* $v_i \in \mathbb{R}^{|S|}$. As we will show (in Section A), for a given training direction $v$, the fraction of variance that $v$ explains[3] in the datamodel representations $\{\theta_x^{(i)}\}$ captures the importance of the corresponding

---

[3]The *fraction of explained variance* of a given vector $v \in \mathbb{R}^d$ in a set of vectors $\{\theta_i \in \mathbb{R}^d\}$ is the empirical variance of $v^\top \theta_i$ divided by the total amount of variance in $\{\theta_i\}$ (i.e., trace($\text{Cov}[\theta_i]$)). In other words, this measures what fraction of the total variation in $\{\theta_i\}$ is along the direction $v$.

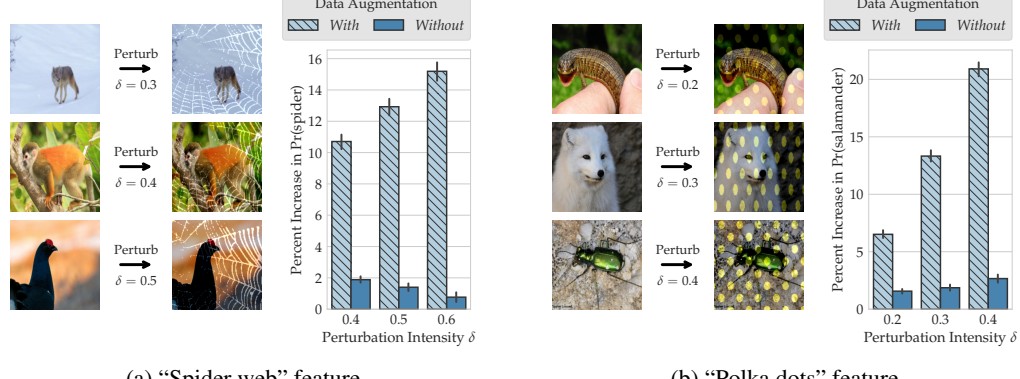

(a) "Spider web" feature          (b) "Polka dots" feature

Figure 4: **Effect of data augmentation on LIVING17 models**. Standard data augmentation amplifies specific instances of co-occurrence (panel (a)) and texture (panel (b)) biases. The left side of each panel illustrates the distinguishing feature transformation at three different levels of intensity $\delta$. On the right, we plot the average treatment effect of the transformation on the predicted confidence of models trained with and without data augmentation, for varying $\delta$. We find that—at moderate intensity $\delta$—adding a spider web pattern to images makes models trained with (without) data augmentation, on average, 13% (1%) more confident in predicting the class "spider," while overlaying a polka dot pattern makes them 14% (2%) more confident in the class "salamander." In both cases, increasing the intensity $\delta$ widens the gap between the two model classes.

combination of training examples to model predictions for algorithm $\mathcal{A}_i$. Thus, we would hope for training directions that distinguish $\mathcal{A}_1$ from $\mathcal{A}_2$ to explain a high (resp., low) amount of the variance in datamodel representations of algorithm $\mathcal{A}_1$ (resp., $\mathcal{A}_2$).

Figure 3 displays the identified distinguishing training directions surfaced in the first stage. On the left, we can see that the training directions distinguishing $\mathcal{A}_1$ from $\mathcal{A}_2$ (in green) indeed explain a significant amount of variance in the datamodels of $\mathcal{A}_1$ but not in those of $\mathcal{A}_2$. Visualizing the subpopulations corresponding to two of the distinguishing directions (Figure 3 right) suggests the following distinguishing features:

- **Spider web:** Direction **A** surfaces a subpopulation of test images that contain spiders. In stark contrast to random images of spiders in LIVING17 data, all the images in the surfaced subpopulation contain a white *spider web* in the background. This leads us to hypothesize that models trained with standard data augmentation—moreso than those trained without it—use spider webs to predict the class "spider". We test this hypothesis using a feature transformation that overlays a spider web pattern onto an entire image (see Figure 4a).

- **Polka dots:** Direction **B** surfaces a subpopulation of test images that contain salamanders. Again, comparing these images to random salamander images in LIVING17 test data reveals a shared property of the surfaced subpopulation, namely, the presence of yellow-black polka dots. This suggests that this texture is a distinguishing feature—models trained with data augmentation strongly rely on the polka dot texture to predict the class "salamander." To test this hypothesis, we design a feature transformation that adds yellow-black polka dots to the entire image (see Figure 4b).

**Findings.** In Figure 4, we compare the effect of the above feature transformations on models trained with and without data augmentation. The results confirm our hypotheses put forth above. In particular, overlaying a spider web pattern with varying (30%/40%/50%) opacity increases $P(\text{"spider"})$ (i.e., models' average softmax confidence in the spider label) predicted in models trained with data augmentation (11%/13%/15% increase) signifantly more than in those without (2%/1%/1%). Similarly, overlaying the yellow polka dot texture with varying opacity (20%/30%/40%) increases $P(\text{"salamander"})$ far more when we use data augmentation (6%/14%/21%) than when we do not (1%/2%/2%). Furthermore, increasing the transformations' intensity consistently widens the gap between the predictions of the two model classes. Overall,

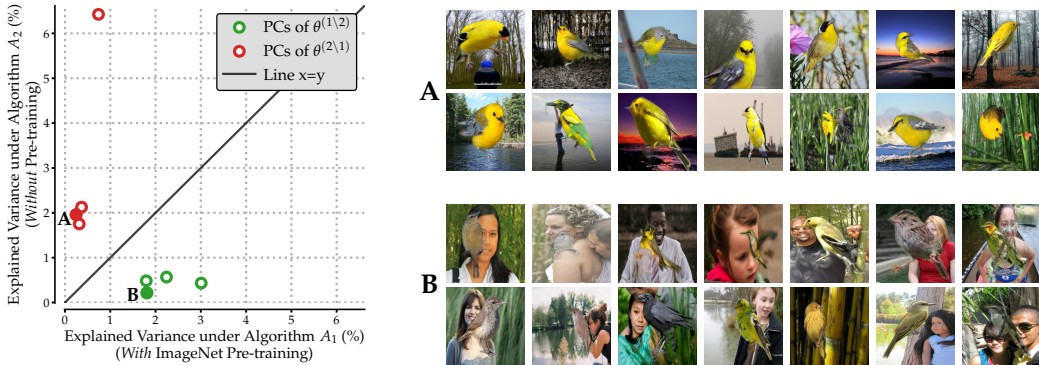

Figure 5: **Comparing Waterbirds models trained with and without ImageNet pre-training.** An analogous figure to Figure 3 for our second case study, see Figure 3 for a description. In this case, direction **A** seems to correspond to yellow birds and direction **B** to human faces in the background.

these differences verify that the feature transformations we constructed are indeed able to distinguish the two learning algorithms as according to Definition 2.

**Connection to previous work.** Our case study demonstrates how training with standard data augmentation on LIVING17 data can amplify specific instances of co-occurrence bias (spider webs) and texture bias (polka dots). These findings are consistent with prior works that show that data augmentation can introduce biases (e.g., (Hermann et al., 2020)). Also, the distinguishing features that we find are specific to certain classes—this may explain why data augmentation can have disparate impacts on performance across different classes (Balestriero et al., 2022). Finally, our findings, which demonstrate how data augmentation alters relative importance of specific features, corroborate the view of data augmentation as *feature manipulation* (Shen et al., 2022).

## 3.2 CASE STUDY: PRE-TRAINING

Pre-training models on large datasets is a standard transfer learning approach to improve performance on downstream tasks where training data is scarce. Our case study focuses the effect of ImageNet pre-training (Deng et al., 2009; Kornblith et al., 2019b) in the context of classifiers trained on the WATERBIRDS dataset (Sagawa et al., 2020). The task is to classify images of birds as "waterbird" or "landbird"—these labels spuriously correlate with "land" and "water" backgrounds respectively. Pre-training significantly improves the "worst group" accuracy on images where the background conflicts with the bird—but how does it impact the fine-grained features learned by the models? To study this, we compare two classes of ResNet-50 models trained with the exact same settings modulo the use of ImageNet pre-training. That is, we consider the following two learning algorithms:

- **Algorithm $\mathcal{A}_1$:** Pre-training on ImageNet, followed by full-network finetuning on Waterbirds. Resulting models attain $89.1\%$ average accuracy and $63.6\%$ worst-group accuracy on the test set.
- **Algorithm $\mathcal{A}_2$:** Training from scratch on Waterbirds data. Corresponding models attain $63.9\%$ average accuracy and $5.7\%$ worst-group accuracy on the test set.

**Identifying distinguishing features.** Applying our framework analogously as done in Section 3.1, we identify two candidate distinguishing features:

- **Yellow color:** Direction **A** surfaces a subpopulation of test images that contain yellow birds belonging to class "landbird." This leads us to hypothesize that models trained from scratch (i.e., without ImageNet pre-training) spuriously rely on the color yellow to predict the class "landbird," whereas ImageNet-pretrained models do not. Additional analysis (Appendix D) supports this hypothesis, as training images that contain other yellow objects strongly influence the predictions of models trained from scratch on this subpopulation. To test this hypothesis, we design a feature transformation that adds a yellow square patch to images (see Figure 6a).

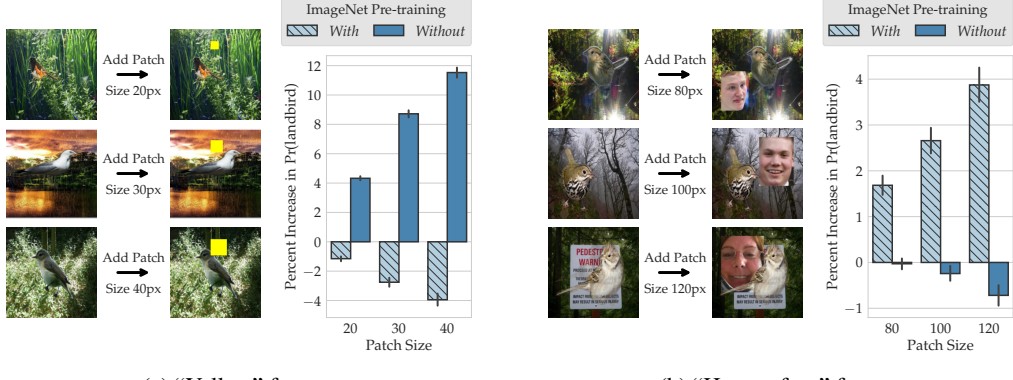

(a) "Yellow" feature                    (b) "Human face" feature

Figure 6: **Effect of ImageNet pre-training on WATERBIRDS classification**. Analogously to Figure 4, we use our framework to identify spurious correlations that are either suppressed or amplified by ImageNet pre-training. **(Left)** Adding a yellow patch to images makes models trained without (with) pre-training, on average, 9% more (2% less) confident in predicting the label "landbird." **(Right)** Adding human faces to image backgrounds makes models trained with (without) pre-training, on average, 3% (0%) more confident in predicting "landbird." In both cases, increasing the transformation intensity widens the gap in treatment effect between the two model classes.

- **Human face:** Direction **B** surfaces a subpopulation of "landbird" that have *human faces* in the background. This suggests that ImageNet pre-training introduces a spurious dependence on human faces to predict the label "landbird." Additional analysis supports this hypothesis, as training images containing face(s) strongly influence the predictions of ImageNet-pretrained models on this subpopulation (see Appendix D). To test this hypothesis, we design a transformation that inserts patches of human faces in WATERBIRDS image backgrounds (see Figure 6b).

**Findings.** In Figure 6, we compare the effect of the above feature transformations on models trained with and without ImageNet pre-training. The results confirm both of our hypotheses. Adding a yellow square patch with varying size (20/30/40 px) to test images increased $P$("landbird") by 4%/9%/12% for models trained from scratch but *decreased* $P$("landbird") for models pre-trained on ImageNet. Similarly, adding a human face patch[4] to image backgrounds increased $P$("landbird") by 2%/3%/4% for pre-trained models, but did not significantly affect models trained from scratch. Once again, increasing the intensity (i.e., patch size) of these feature transformations further widens the gap in sensitivity between the two model classes.

**Connections to prior work.** This case study pinpoints how pre-training can alter the importance of different spurious correlations. In particular, our results show that ImageNet pre-training reduces dependence on some spurious correlations (e.g., yellow color $\rightarrow$ landbird) but also *introduces* new ones (e.g., human face $\rightarrow$ landbird). Our findings thus shed light on two seemingly contradictory phenomena: pre-training can *simultaneously* improve robustness to spurious features (Ghosal et al., 2022; Tu et al., 2020) in target data as well as transfer new spurious correlations (Salman et al., 2022; Neyshabur et al., 2020) from the pre-training dataset.

### 3.3 CASE STUDY: SGD HYPERPARAMETERS

The choices of optimizer and corresponding hyperparameters can affect both the trainability and the generalization of resulting models (Hoffer et al., 2017). In this case study, we focus on stochastic gradient descent (SGD), and its hyperparameters—learning rate and batch size—that control the effective scale of the noise in SGD. We defer our findings to Appendix B due to space constraints.

---

[4]We collect a bank of human face patches using ImageNet validation examples and their corresponding face annotation (Yang et al., 2022); see Appendix C.4 for more information.

## 4 RELATED WORK

We now compare and contrast our approach to algorithm comparison with approaches to the related problem of *model comparison*, where one tries to characterize the difference between two (usually fixed) machine learning models. A long line of work has sought to design methods for characterizing the differences between models:

**Representation-based comparison.** A popular approach (particularly in the context of deep learning) is to compare models using their internal *representations*. Since the coordinates of these representations do not have a consistent interpretation, representation-based model comparison typically studies the degree to which different models' representations can be aligned. Methods based this approach include canonical correlation analysis (CCA) and variants (Raghu et al., 2017; Morcos et al., 2018a; Cui et al., 2022), centered kernel alignment (CKA) (Kornblith et al., 2019a), graph-based methods (Li et al., 2015; Chen et al., 2021), and model stitching (Csiszarik et al., 2021; Bansal et al., 2021). Prior works have used these methods to compare wide and deep neural networks (Nguyen et al., 2021); vision transformers and convolutional networks (Raghu et al., 2021); pre-trained and trained-from-scratch models (Neyshabur et al., 2020); and different language models (Wu et al., 2020). Though they are often useful, prior work shows that representation-based similarity measures are not always reliable for testing functional differences in models (Ding et al., 2021). Our approach to algorithm comparison differs from these methods in both objective and implementation:

*Learning algorithms, not fixed models*: Rather than focusing on a single fixed model, our objective here is to compare the class of models that result from a given learning algorithm. In particular, we aim to find only differences that arise from algorithmic design choices, and not those that arise from the (sometimes significant) variability in training across random seeds (Zhong et al., 2021).

*Feature-based, not similarity-based*: Methods such as CCA and CKA focus on outputting a single score that reflects the overall similarity between two models. On the other hand, the goal of our framework is to find fine-grained differences in model behavior. Still, in Appendix F.1 we show that we can also use our method for more global comparisons, for instance by computing the average cosine similarity of the datamodel vectors.

*Model-agnostic*: Our framework is agnostic to type of model used and thus allows one to easily compare models across learning algorithms—our method extends even to learning algorithms that do not have explicit representations (e.g., decision trees and kernel methods).

**Example-level comparisons.** An alternative method for comparing models is to compare their predictions directly. For example, Zhong et al. (2021) compare predictions of small and large language models (on a per-example level) to find that larger models are not uniformly better across example. Similarly, Mania et al. (2019) study the *agreement* between models, i.e., how often they output the same prediction on a per-example level. In another vein, Meding et al. (2022) show that after removing impossible or trivial examples from test sets, different models exhibit more variations in their predictions. Our framework also studies instance-level predictions, but ultimately connects the results back to human-interpretable distinguishing features.

**Comparing feature attributions.** Finally, another line of work compares models in terms of how they use features at test time. In the presence of a known set of features, one can compute feature importances and compare them across models (Wang et al., 2022). In cases where we do not have access to high-level features, we can use instance-level explanation methods such as saliency maps to highlight different parts of the input, but these methods generally do not help at distinguishing models (Denain & Steinhardt, 2022). Furthermore, multiple evaluation metrics (Adebayo et al., 2018; Hooker et al., 2018; Shah et al., 2021) indicate that common instance-specific feature attribution methods can fail at accurately highlighting features learned by the model.

## 5 CONCLUSION

We introduce a unified framework for fine-grained comparisons of any two learning algorithms. Specifically, our framework compares models trained using two different algorithms in terms of how the models *rely on training data* to make predictions. Through three case studies, we showcase the utility of our framework in pinpointing how three aspects of the standard training pipeline—data augmentation, pre-training, optimization—can shape model behavior.

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

# Appendices

# A    ALGORITHM ANALYSIS

In the main paper, we applied our comparison framework to identify feature transformations that distinguished three pairs of learning algorithms. Here, we describe in more detail the algorithmic stage of that framework, i.e., the stage whose purpose is to find *distinguishing training directions*. To this end, we walk through each of the three steps of the algorithm presented in Section 2.2 and provide intuition for how they identify distinguishing training directions.

## A.1    A PRIMER ON DATAMODEL REPRESENTATIONS

The first step in our method is to compute datamodel vectors $\theta_j^{(i)} \in \mathbb{R}^{|S|}$, one for each test input $x_j$. Each datamodel vector encodes the importance of individual training examples $S$ to model's loss at input $x_j$ when trained with learning algorithm $\mathcal{A}_i$. More specifically, each vector corresponds to the solution to a specific regression problem—these regression problems (explained below) form the basis of our analysis.

**Setting up the regression problem.** Let us fix a single learning algorithm $\mathcal{A}$ (being $\mathcal{A}_1$ or $\mathcal{A}_2$). For a given training set $S = \{x_1, \ldots, x_d\}$, a test input $x$, and subset $S' \subset S$ of the training set we define the *model output function* as:

$$f(x, S') = \text{ the loss after training a model on } S' \text{ and evaluating on } x.$$

For example, $f(x, S')$ can encode training a deep neural network on the subset $S'$, then computing the network's *correct-class margin* on the input $x$. For a fixed $x$, the corresponding regression problem is to predict the model output $f(x, S')$ given a subset $S'$.

**Datamodels.** Ilyas et al. (2022) show that—for deep neural networks trained on standard image classification tasks—we can solve the regression problem above with a simple *linear* predictor. More specifically, they showed that

$$\mathbb{E}[f(x, S')] \approx \theta_x \cdot \mathbf{1}_{S'}, \tag{1}$$

where $\theta_x$ is a (learned) parameter vector (called the *datamodel* for $x$), and $\mathbf{1}_{S'} \in \{0, 1\}^{|S|}$ is a binary *indicator vector* of the set $S'$, encoding whether each example of $S$ is included in $S'$, i.e.,

$$(\mathbf{1}_{S'})_i = \begin{cases} 1 & \text{if } x_i \in S' \\ 0 & \text{otherwise.} \end{cases}$$

In this way, datamodels consistitute linear approximations of model output functions.

**Datamodels as a representation space.** While each datamodel is specific to an individual test input, we can treat a collection of datamodels as *embeddings or representations* of test inputs $\{x_j\}_j$ into a common $|S|$-dimensional space. By comparing the representations, we can analyze the structure of the data—as used by the specific learning algorithm under study. Furthermore, these representations have a number of properties that make them useful for algorithm comparisons:

(a) **Consistent basis:** A datamodel representation for a fixed train set $S$ always has the same basis: coordinate $i$ corresponds to the importance of the $i$-th training example. This consistency makes datamodels a convenient medium for algorithm comparisons, as representations are *automatically aligned* across different learning algorithms, and even across models that lack explicit representations (e.g., decision trees).

(b) **Predictiveness:** Datamodel vectors are *causally predictive* of model behavior. That is, as Ilyas et al. (2022) show, we can use them to predict the counterfactual impact of removing or adding different training examples on model output for a given test example. As a result, any trends we find across the datamodel representations come with a precise quantitative interpretation in terms of model outputs (to which we will come back to later in thise section).

(c) **Density:** Datamodel representations also have a *high effective dimensionality*: that is, one needs thousands of components to explain significant fraction of variance, for instance, on CIFAR-10 Ilyas et al. (2022).[5] This suggests that datamodel representations encode fine-grained information about how each learning algorithm uses the training data, making them useful for uncovering subtle differences in model behavior.

---

[5]This is in stark contrast to "standard" representations derived from the penultimate layer of a trained model, which tend to have effective dimensions that are much lower, typically equal to number of classes minus one.

## A.2 Residual datamodels

In Step 2 of our algorithm, using the two sets of datamodels $\{\theta^{(1)}\}$ and $\{\theta^{(2)}\}$, we compute the residual datamodel vectors:

$$\theta_i^{(1\backslash 2)} = \theta_i^{(1)} - \langle \theta_i^{(1)}, \theta_i^{(2)} \rangle \, \theta_i^{(2)},$$

(Note that this operation only makes sense because the two sets of datamodels live in the same vector space—see property (a) above.) As we demonstrate, they correspond to datamodels for a certain *residual model output* function that we define below.

Recall that our overarching goal is to find training directions that that strongly influence models trained with learning algorithm $\mathcal{A}_1$ but not $\mathcal{A}_2$ (or vice-versa) when classifying $x$. More specifically, we want to find training directions $u$ that strongly influences $f^{(1)}(x, S)$, while ignoring directions that also influence $f^{(2)}(x, S)$. In other words, what we care about is the "residual" of $f^{(1)}$ after removing the part that is *correlated*[6] with $f^{(2)}$. To capture this, consider the *residual model output* function of $\mathcal{A}_1$ relative to $\mathcal{A}_2$:

$$f^{(1\backslash 2)}(x, S) := f^{(1)}(x, S) - \rho_{f^{(1)} f^{(2)}} \cdot f^{(2)}(x, S)$$

where $\rho_{f^{(1)} f^{(2)}}$ is the correlation between two model output functions (across varying $S$). It turns out that the datamodel for the residual output function is given precisely by the *residual datamodel* defined in Step 2 of our algorithm: the projection of one (normalized) datamodel representation into the nullspace of the other representation. That is, residual datamodels correspond to a linear approximation of the residual model output:

$$\mathbb{E}[f^{(1\backslash 2)}(x, S)] \approx \theta^{(1\backslash 2)} \cdot \mathbf{1}_S$$

In summary, Step 2 of our procedure reduces understanding the differences in learning algorithms $\mathcal{A}_1$ and $\mathcal{A}_2$ to analyzing their residual model outputs via residual datamodels. The residual datamodels highlight directions that influence learning algorithm $\mathcal{A}_1$ after *projecting away* directions that also influence learning algorithm $\mathcal{A}_2$.

## A.3 Finding global trends with PCA

The output of Step 2 of our procedure is two set of residual datamodels—one set looking at the residual of $\mathcal{A}_1$ with respect to $\mathcal{A}_2$, and vice versa. These residual datamodels capture directions that each algorithm is most sensitive to, though still on a *per-example* level. On the other hand, our goal is to find directions that each algorithm is most sensitive to on an *aggregate* level (as in Definition 2). We now show how the final step in our procedure, computing PCA on the residual datamodel representations, achieves this goal.

For a given input $x$ with datamodel $\theta$, and for a training direction $u \in \mathbb{R}^{|S|}$, we can estimate the example's sensitivity to $u$ as $[(\theta \cdot u)]^2$—that is, the (estimated) effect on $f(x, S)$ of upweighting and downweighting the training samples according to $u$. Since our goal is to find a direction that the residual model output function $f^{(1\backslash 2)}$ is most sensitive *in aggregate*, our objective is

$$\arg \max_{u \in \mathbb{S}^{n-1}} \mathbb{E}_{x \in T}[(\theta_x^{(1\backslash 2)} \cdot u)^2].$$

The direction that maximizes this objective is exactly the dominant principal component of the set of residual datamodels! Similarly, the top $k$ principal components correspond to directions that algorithm $\mathcal{A}_1$ is most sensitive to after accounting for $\mathcal{A}_2$.

In summary, Step 3 of our procedure finds directions that strongly influence only one learning algorithm by examining directions of highest explained variance of the residual datamodel vectors.

---

[6]Over the distribution of $S$.

# B  CASE STUDY: SGD HYPERPARAMETERS

The choices of optimizer and corresponding hyperparameters can affect both the trainability and the generalization of resulting models (Hoffer et al., 2017; Keskar et al., 2017). In this case study, we focus on stochastic gradient descent (SGD), and its hyperparameters—learning rate and batch size—that control the effective scale of the noise in SGD. We study the effect of these hyperparameters in the context of CIFAR-10 (Krizhevsky, 2009) classifiers by comparing the following two learning algorithms:

- **Algorithm $\mathcal{A}_1$:** Training with high SGD noise: large learning rate (0.1) and small batch size (256). Resulting models attain 93% average test accuracy.

- **Algorithm $\mathcal{A}_2$:** Training with low SGD noise: small large rate (0.02) and large batch size (1024). Resulting models attain 89% average test accuracy.

**Identifying distinguishing features.** Applying our framework analogously as done in previous case studies, we identify two candidate distinguishing features:

- **Black-and-white texture:** Direction **A** surfaces a subpopulation of black-and-white dogs (Figure 5, top right). Additional analysis in Appendix D shows that a subset of training images with black-and-white objects (e.g., ships) influence predictions on this subpopulation only when models are trained with low SGD noise (algorithm $\mathcal{A}_2$). This leads us to hypothesize that models trained with low SGD noise rely more on black-and-white textural features to predict the class "dog." To test this hypothesis, we design a feature transformation that modifies a given image with a small black-and-white patch, which loosely resembles the face and nose of dogs in the surfaced subpopulation (see Figure 8a).

- **Rectangular shape:** Direction **B** surfaces a subpopulation of front-facing trucks (Figure 5, bottom right) with a shared characteristic: rectangular-shaped cabin and cargo area. Additional analysis in Appendix D shows that a subset of training images with rectangular patterns influence predictions on this subpopulation only when models trained with low SGD noise. This suggests that models trained with low SGD noise partially rely on rectangular-shaped patterns to predict the class "truck". To test this hypothesis, our feature transformation modifies a given image with a patch of high-contrast rectangles, which loosely resembles the cabin / cargo shape of trucks in the surfaced subpopulation (see Figure 8b).

**Findings.** In Figure 8, we compare the effect of the above feature transformations on models trained with high and low SGD noise. The results again confirm both hypotheses. Adding a black-and-white patch of varying size (4/5/6 px) to test images increased $P$("dog") by 8/12/14% for models trained with low SGD noise, but only by 6/8/9% for models trained with high SGD noise. Similarly, applying a rectangular-shape patch of varying size (6/7/8 px) increased $P$("truck") by 4/6/7% for models trained with low SGD noise, but only by 1/2/2% for models trained with high SGD noise. Once again, increasing the intensity (i.e., patch size) of these feature transformations widens the gap in sensitivity between the two model classes.

**Connections to prior work.** This case study shows how reducing the scale of SGD noise can increase reliance on certain low-level features (e.g., rectangular shape $\rightarrow$ trucks). While prior works show that lower SGD noise worsens aggregate model performance (Keskar et al., 2017; Wen et al., 2019), our methodology identifies *specific* features that are amplified due to low SGD noise. Furthermore, the simplistic nature of the identified features corroborate the theoretical results put forth in Li et al. (2019): the learning rate scale determines the extent to which models memorize patterns that are easy-to-fit but hard-to-generalize. More broadly, our framework motivates a closer look at how features amplified via low SGD noise collectively alter aggregate model performance.

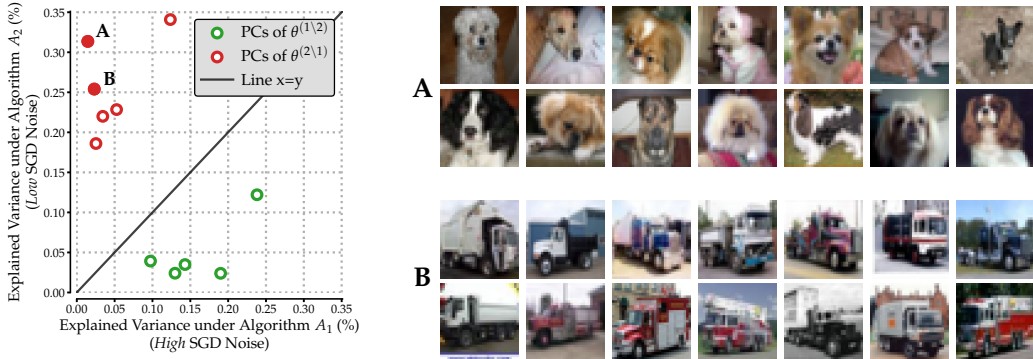

Figure 7: **Comparing CIFAR-10 models trained with high and low SGD noise.** An analogous figure to Figure 3 for our third case study, see Figure 3 for a description. In this case, direction **A** seems to correspond to dogs with a particular black-and-white texture, and direction **B** to front-facing trucks with prominent rectangular component(s).

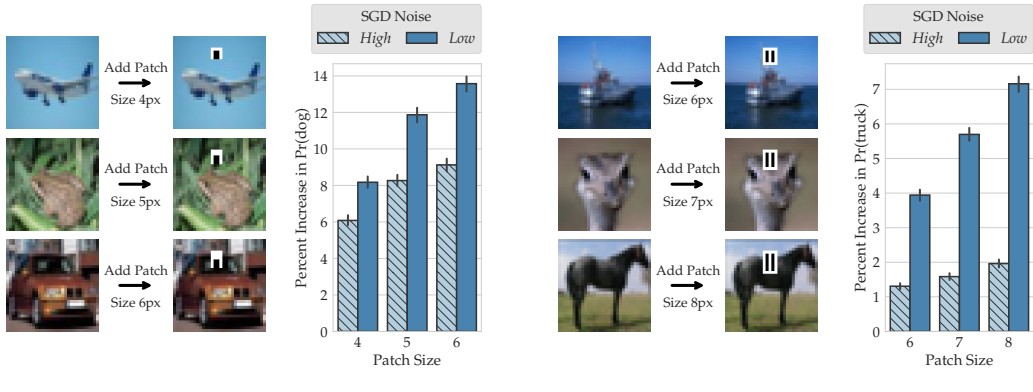

(a) "Black-and-white texture" feature                    (b) "Rectangular shape" feature

Figure 8: **Effect of SGD hyperparameters on CIFAR-10 models**. Analogously to Figures 4 and 6, we use our framework to identify features that distinguish models trained with lower SGD noise from models trained with higher SGD noise. **(Left)** Adding a black-and-white patch to images makes models trained with low (high) SGD noise, on average, 11% (8%) more confident in predicting the label "dog." **(Right)** Adding high-contrast rectangles to images makes models trained with low (high) SGD noise, on average, 5.5% (1.5%) more confident in predicting the label "truck." In both cases, increasing the intensity of feature transformations widens the gap in treatment effect between the two model classes.

## C    EXPERIMENTAL SETUP

In this section, we outline the experimental setup—datasets, models, training algorithms, hyperparameters, and datmodels—used for our case studies in Section 3.

### C.1    DATASETS

**Living17.** The Living17 dataset (Santurkar et al., 2021) is an ImageNet-derived dataset, where the task is to classify images belonging to 17 types of living organisms (e.g., salamander, bear, fox). Each Living17 class corresponds to an ImageNet superclass (i.e., a set of ImageNet classes aggregated using WordNet (Miller, 1995)). Santurkar et al. (2021) introduce Living17 as one of four benchmark to evaluate model robustness to realistic subpopulation shifts. In our case study, we study the effect of data augmentation using a variant of this dataset, wherein the training and test images belong to the same set of subpopulations (i.e., no subpopulation shift).

**Waterbirds.** The Waterbirds dataset (Sagawa et al., 2020) consists of bird images taken from the CUB dataset (Wah et al., 2011) and pasted on backgrounds from the Places dataset (Zhou et al., 2017). The task here is to classify "waterbirds" and "landbirds" in the presence of spurious correlated "land" and "water" backgrounds in the training data. Sagawa et al. (2020) introduce Waterbirds as a benchmark to evaluate models under subpopulation shifts induced by spurious correlations. In our case study, we compare how models trained from scratch on Waterbirds data differ from ImageNet-pretrained models that are fine-tuned on Waterbirds data.

**CIFAR-10.** We consider the standard CIFAR-10 (Krizhevsky, 2009) image classification dataset in order to study the effect of two SGD hyperparameters: learning rate and batch size.

Summary statistics of the datasets described above are outlined in Table 1.

Table 1: Summary statistics of datasets

| Dataset | Classes | Size (Train/Test) | Input Dimensions |
|---|---|---|---|
| Living17 | 17 | 88,400/3,400 | $3 \times 224 \times 224$ |
| Waterbirds | 2 | 4,795/5,794 | $3 \times 224 \times 224$ |
| CIFAR-10 | 10 | 50,000/10,000 | $3 \times 32 \times 32$ |

### C.2    MODELS, LEARNING ALGORITHMS, AND HYPERPARAMETERS

**Living17.** We use the standard ResNet18 architecture (He et al., 2015) from the `torchvision` library. We train models for 25 epochs using SGD with the following configuration: initial learning rate 0.6, batch size 1024, cyclic learning rate schedule (with peak at epoch 12), momentum 0.9, weight decay 0.0005, and label smoothing (with smoothing hyperparameter 0.1). To study the effect of data augmentation, we train models with the following algorithms:

- **Algorithm $\mathcal{A}_1$ (with data augmentation)**: Models are trained with standard data augmentation: random resized cropping (with default `torchvision` hyperparamters) and random horizontal flips. On average, models attain 89.2% average test accuracy.

- **Algorithm $\mathcal{A}_2$ (without data augmentation)**: Models are trained without data augmentation. On average, models attain 81.9% average test accuracy.

**Waterbirds.** We use the standard ResNet50 architecture from the `torchvision` library. We train models using SGD with momentum 0.9 and weight decay 0.0001 for a maximum of 50 epochs (and stop early if the training loss drops below 0.01). For model selection, we choose the model checkpoint that has the maximum average accuracy on the validation dataset. As in Sagawa et al. (2020), we do not use data augmentation. In our case study on pre-training, we consider ImageNet pretrained models from `torchvision`. We consider models trained using the following algorithms:

- **Algorithm $\mathcal{A}_1$ (ImageNet pre-training)**: Models pre-trained on ImageNet are fully fine-tuned on Waterbirds data with a fixed SGD learning rate 0.005 and batch size 64. On average, models attain 89.1% (non-adjusted) average test accuracy and 63.9% worst-group test accuracy.

- **Algorithm $\mathcal{A}_2$ (Training from scratch)**: Models are trained from scratch (i.e., random initialization) on Waterbirds data with SGD: initial learning rate $0.01$, batch size $64$, and a linear learning rate schedule ($0.2\times$ every 15 epochs). On average, models attain $63.6\%$ average test accuracy and $5.7\%$ worst-group test accuracy.

**CIFAR-10.** We use the ResNet9 architecture from Kakao Brain[7], which is optimized for fast training. We train models using SGD with momentum $0.9$ and weight decay $0.0005$ for a maximum of 100 epochs (and stop early if the training loss drops below $0.01$). We augment training data with a standard data augmentation scheme: random resized cropping with 4px padding and random horizontal flips. To study the effect of SGD noise in our case study, we vary learning rate and batch size. Specifically, we compare models trained with the following algorithms:

- **Algorithm $\mathcal{A}_1$ (high SGD noise)**: Models are trained with SGD using a large initial learning rate ($0.1$), small batch size ($256$), and a linear learning rate schedule ($0.5\times$ every 20 epochs). On average, models attain $93.3\%$ test accuracy.

- **Algorithm $\mathcal{A}_2$ (low SGD noise)**: Models are trained with SGD using a small fixed learning rate ($0.02$) and large batch size ($1024$). On average, models attain $89.5\%$ test accuracy.

## C.3 DATAMODELS

Now, we provide additional details on datamodels which, we recall, are used in the first stage of our algorithm comparison framework (see Section 2).

**Estimating linear datamodels.** Recall from Appendix A that the datamodel vector for example $x_j$, $\theta_j^{(i)} \in \mathbb{R}^{|S|}$, encodes the importance of individual training examples $S$ to model's loss at example $x_j$ when trained with algorithm $\mathcal{A}_i$. Concretely, given test example $x_j$ and training set $S = \{x_1, \ldots, x_d\}$, the datamodel $\theta_j$ is a sparse linear model (or surrogate function) trained on the following regression task: For a training subset $S' \subset S$, can we predict the correct-class margin $f_{\mathcal{A}}(x_j; S')$ of a model trained on $S'$ with algorithm $\mathcal{A}$? This task can be naturally formulated as the following supervised learning problem: Given a training set $\{(S_i, f_{\mathcal{A}}(x; S_i))\}_{i=1}^{m}$ of size $m$, the datamodel $\theta_j$ (for example $x_j$) is the solution to the following problem:

$$\theta_j = \min_{w \in \mathbb{R}^{|S|}} \frac{1}{m} \sum_{i=1}^{m} \left( w^{\top} \mathbf{1}_{S_i} - f_{\mathcal{A}}(x_j; S_i) \right)^2 + \lambda \|w\|_1, \tag{2}$$

where $\mathbf{1}_{S_i}$ is a boolean vector that indicates whether examples in the training dataset $x \in S$ belong to the training subset $S_i$. Note that each datamodel training point $(S_i, f_{\mathcal{A}}(x_j, S_i))$ is obtained by (a) training a model $f$ (e.g., ResNet9) on a subset of data $S_i$ (e.g., randomly subsampled CIFAR data) and (b) evaluating the trained model's output on example $x_j$. Ilyas et al. (2022) demonstrate that linear datamodels can accurately (and counterfactually) predict outputs of deep image classifiers.

**Datamodel estimation hyperparameters.** Recall that our algorithm comparison framework in Section 2 involves estimating two sets of datamodels $\{\theta^{(1)}\}$ and $\{\theta^{(2)}\}$ for learning algorithms $\mathcal{A}_1$ and $\mathcal{A}_2$ respectively. In our case studies, we estimate two datamodels, $\theta_i^{(1)}$ and $\theta_i^{(2)}$ for every example $x_i$ in the test dataset. Estimating these datamodels entail three design choices:

- **Sampling scheme for train subsets:** Like in Ilyas et al. (2022), we use $\alpha$-random subsets of the training data, where $\alpha$ denotes the subsampling fraction; we set $\alpha = 50\%$ as it maximizes sample efficiency (or model reuse) for empirical influence estimation (Feldman & Zhang, 2020), which is equivalent to a variant of linear datamodels.

- **Sample size for datamodel estimation:** Recall that a datamodel training set of size $m$ corresponds to training $m$ models (e.g., $m$ ResNet18 models on CIFAR-10) on independently sampled train subsets (or masks). We estimate datamodels on LIVING17, WATERBIRDS, and CIFAR-10 using $120k$, $50k$, and $50k$ samples (or models) per learning algorithm respectively; we make a validation split using $10\%$ of these samples.

---

[7]https://github.com/wbaek/torchskeleton/blob/master/bin/dawnbench/cifar10.py

- $\ell_1$ **sparsity regularization:** We use cross-validation to select the sparsity regularization parameter $\lambda$. Specifically, for each datamodel, we evaluate the MSE on a validation split to search over $k = 50$ logarithmically spaced values for $\lambda$ along the regularization path. We then re-compute the datamodel on the entire dataset with the optimal $\lambda$ value and all $m$ training examples.

We provide pseudocode for estimating datamodels in Algorithm 1.

---

**Algorithm 1** An outline of the datamodeling framework introduced in Ilyas et al. (2022).

---

1: **procedure** ESTIMATEDATAMODEL(target example $x$, train set $S$ of size $d$, subsampling frac. $\alpha \in (0,1)$)
2:     $T \leftarrow []$                                                ▷ Initialize *datamodel training set*
3:     **for** $i \in \{1, \ldots, m\}$ **do**
4:         Sample a subset $S_i \subset S$ from $\mathcal{D}_S$ where $|S_i| = \alpha \cdot d$
5:         $y_i \leftarrow f_{\mathcal{A}}(x; S_i)$                  ▷ Train a model on $S_i$ using $\mathcal{A}$, evaluate on $x$
6:         Define $\mathbf{1}_{S_i} \in \{0,1\}^d$ as $(\mathbf{1}_{S_i})_j = 1$ if $x_j \in S_i$ else 0
7:         $T \leftarrow T + [(\mathbf{1}_{S_i}, y_i)]$             ▷ Update datamodel training set
8:     $\theta \leftarrow$ RUNREGRESSION(T)          ▷ Predict the $y_i$ from the $\mathbf{1}_{S_i}$ vectors
9:     **return** $\theta$                       ▷ Result: a weight vector $\theta \in \mathbb{R}^d$

---

### C.4 FEATURE TRANSFORMATIONS

As discussed in Section 2, we counterfactually verify distinguishing features (inferred via human-in-the-loop analysis) by evaluating whether feature transformations change model behavior as hypothesized. Here, we describe the feature transformations used in Section 3 in more detail[8].

**Designing feature transformations.** We design feature transformations that modify examples by adding a specific patch or perturbation. We vary the intensity of patch-based and perturbation-based transformations via patch size $k$ and perturbation intensity $\delta$ respectively. Additional details specific to each case study:

- **Pre-training.** We use patch-based transformations in this case. For the yellow color feature, we add a $k \times k$ square yellow patch to the input. For the human face feature, we add a $k \times k$ image of a human face to the input. To avoid occlusion with objects in the image foreground, we add the human face patch to the background. We make a bank of roughly 300 human faces using ImageNet face annotations (Yang et al., 2022) by (a) cropping out human faces from ImageNet validation examples and (b) manually removing mislabeled, low-resolution, and unclear human face images.
- **Data augmentation**. We design perturbation-based transformations to verify the identified distinguishing features: spider web and polka dots. In both cases, we $\delta$-perturb each input with a random crop of a fixed grayscale spider web or yellow polka dot pattern.
- **SGD hyperparameters.** We use patch-based transformations in this case study. For the black-white texture feature, we add a $k$-sized patch that loosely resembles a black-white dog nose. Similarly, for the rectangular shape feature, we add a $k$-sized patch that loosely resembles windows.

**Evaluating feature transformations.** As shown in Section 3, given two learning algorithms $\mathcal{A}_1$ and $\mathcal{A}_2$, we evaluate whether a feature transformation $F$ changes predictions of models trained with $\mathcal{A}_1$ and $\mathcal{A}_2$ as hypothesized. To evaluate the counterfactual effect of transformation $F$ on model $M$, we evaluate the extent to which applying $F$ to input examples $x$ increases the confidence of models in a particular class $y$. In our experiments, we estimate this counterfactual effect by averaging over all test examples and over 500 models trained with each learning algorithm.

### C.5 TRAINING INFRASTRUCTURE

**Data loading.** We use FFCV[9] (Leclerc et al., 2022), which removes the data loading bottleneck for smaller models, gives a 3-4× improvement in throughput (i.e., number of models a day per GPU).

---

[8]The code for these feature transformations is available at `anonymized-url`.
[9]Webpage: `http://ffcv.io`

**Datamodels regression.** In addition to FFCV, we use the fast-l1 package[10]—a SAGA-based GPU solver for $\ell_1$-regularized regression—to parallelise datamodel estimation.

**Computing resources.** We train our models on a cluster of machines, each with 9 NVIDIA A100 or V100 GPUs and 96 CPU cores. We also use half-precision to increase training speed.

---

[10]Github repository: https://github.com/MadryLab/fast_l1

# D  ADDITIONAL HUMAN-IN-THE-LOOP ANALYSIS

As outlined in Section 2, the second stage of our framework applies human-in-the-loop analysis to infer distinguishing feature transformations from training directions extracted via PCA on residual datamodels. In this section, we present additional human-in-the-loop analysis in order to substantiate the distinguishing features inferred in each case study.

## D.1  TOOLS FOR INFERRING DISTINGUISHING FEATURES FROM PCA SUBPOPULATIONS

In this section, we outline additional human-in-the-loop tools that we use to analyze subpopulations surfaced by principal components (PCs) of residual datamodels.

- **Class-specific visual inspection.** As shown in Section 3, the subpopulation of test examples whose datamodels have maximum projection onto PCs of residual datamodels largely belong to same class. So, a simple-yet-effective way to identify *subpopulation-specific* distinguishing feature(s) is to just visually contrast the surfaced subpopulation from a set of randomly sampled examples that belong to the same class.

- **Relative influence of training examples.** Given a subset of test examples $S' \subset S$, can we identify a set of training examples $T' \subset T$ that strongly influence predictions on $S'$ when models are trained with algorithm $\mathcal{A}_1$ but not when trained with $\mathcal{A}_2$? Given datamodel representations $\{\theta_i^{(1)}\}$ for $\mathcal{A}_1$ and $\{\theta_i^{(1)}\}$ for $\mathcal{A}_2$, we apply a two-step heuristic approach identify training examples with high influence on $\mathcal{A}_1$ relative to $\mathcal{A}_2$:

  - First, given learning algorithm $\mathcal{A}_i$ and test subset $S'$, we estimate the aggregate (positive or negative) influence of training example $x_k$ on subset $S$ by taking the absolute sum over the corresponding datamodel weights: $\sum_{j \in S'} |\theta_{jk}^{(i)}|$.
  - Then, we take the absolute difference between the aggregate influence estimates of training example $x_k$ using $\theta^{(1)}$ and $\theta^{(2)}$. This difference measures the *relative influence* of training example $x_k$ on predictions of test subset $S$ when models are trained with algorithm $\mathcal{A}_1$ instead of algorithm $\mathcal{A}_2$.

In our analysis, we (a) identify training examples that have top-most relative influence estimates and then (b) visually contrast the subsets of test examples (one for each learning algorithm) that are most influenced by these training examples.

### D.2 CASE STUDY: STANDARD DATA AUGMENTATION

Our case study on LIVING17 data in Section 3.1 shows that standard data augmentation can amplify co-occurrence bias (spider web) and texture bias (polka dots). We further substantiate these findings with relative influence analysis (Figure 9) and class-specific visual inspection (Figure 12).

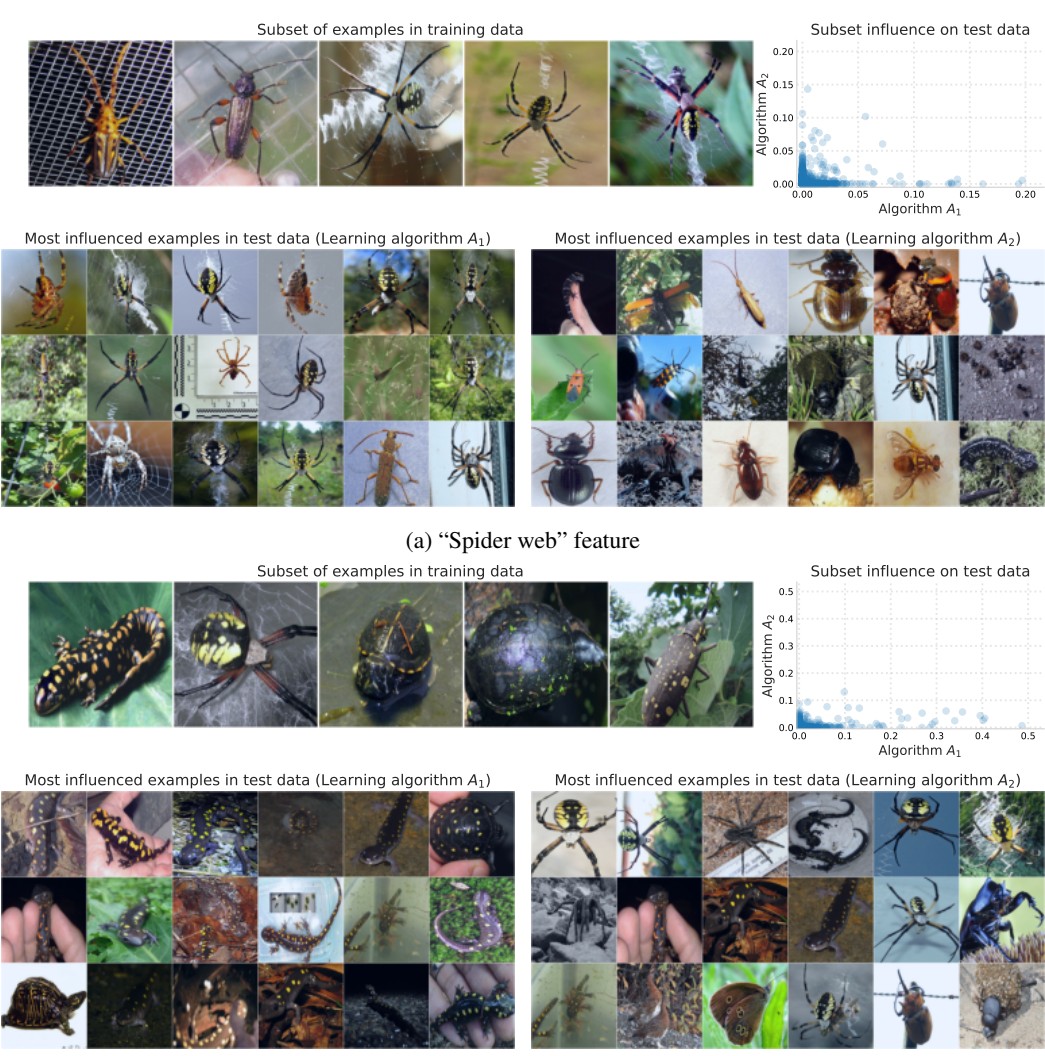

Figure 9: **Relative influence of training data on LIVING17 subpopulations**. **Panel (a)**: Training images that contain web-like patterns have high relative influence on the "spider web" test subpopulation (see Figure 3). These images strongly influence model predictions on test images that contain spider webs (in bottom row) only when models are trained with augmentation (algorithm $\mathcal{A}_1$). **Panel (b)**: Training images that contain yellow-black texture have high relative influence on the "polka dots" test subpopulation (see Figure 3). These images strongly influence model predictions on test images of salamanders with yellow polka dots (in bottom row) only when models are trained with augmentation (algorithm $\mathcal{A}_1$).

### D.3 CASE STUDY: IMAGENET PRE-TRAINING

Our case study on WATERBIRDS data shows that ImageNet pre-training reduces dependence on the "yellow color" feature, but introduces dependence the "human face" feature. We support these findings with relative influence analysis in Figure 10 and additional visual inspection in Figure 13.

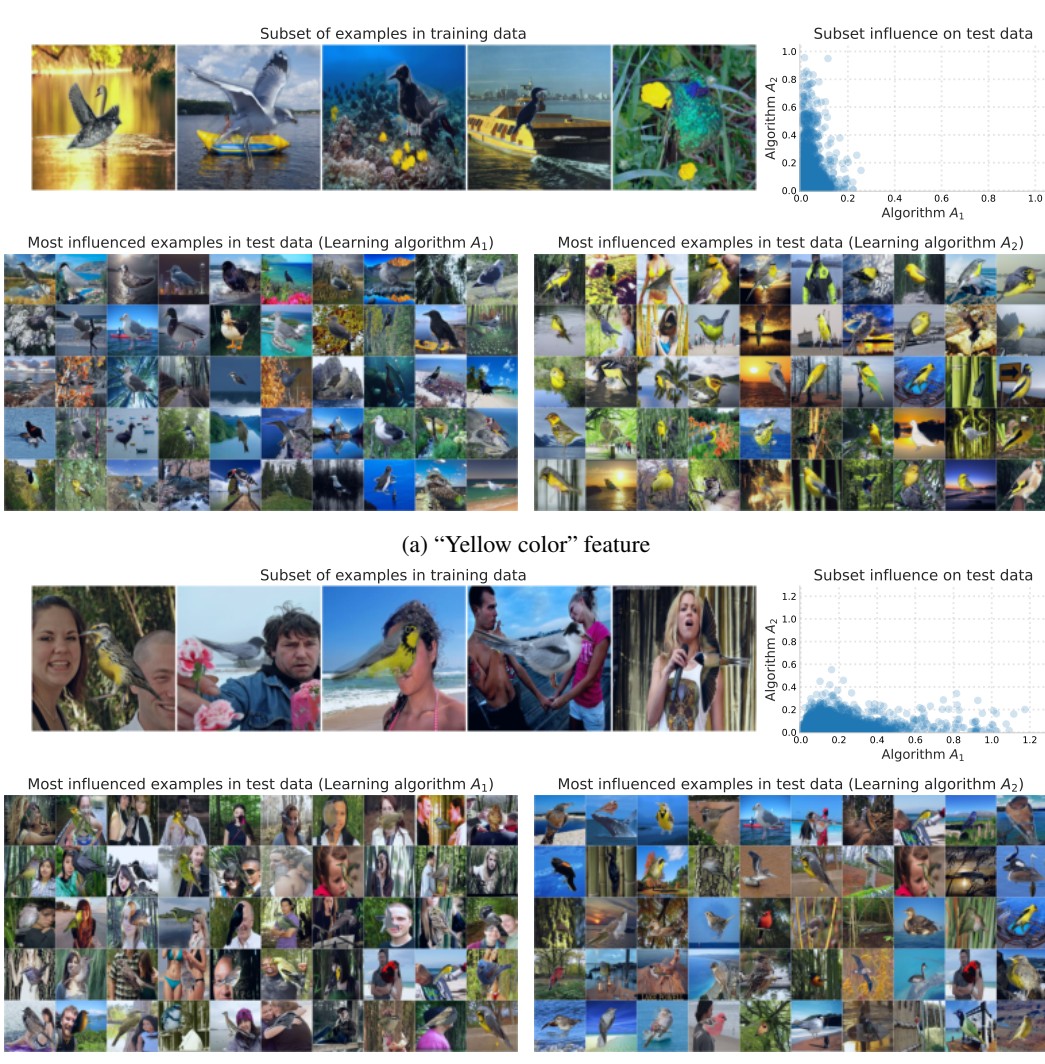

Figure 10: **Relative influence of training data on WATERBIRDS subpopulations**. **Panel (a)**: Training images with yellow objects in the background have high relative influence on the "yellow color" test subpopulation (see Figure 5). These images strongly influence model predictions on test images that have yellow birds / objects (bottom row) only when models are trained from scratch (algorithm $\mathcal{A}_2$). **Panel (b)**: Training images that contain human faces in the background have high relative influence on the "human face" test subpopulation (see Figure 5). These images strongly influence model predictions on test images (in bottom row) with human face(s) only when models are pre-trained on ImageNet (algorithm $\mathcal{A}_1$).

## D.4   CASE STUDY: SGD NOISE

We analyze relative influence (Figure 11), and class-specific subpopulations (Figure 14) to hone in on two instances of distinguishing features–black-and-white texture and rectangular shape—in CIFAR-10 data that are amplified by low SGD noise.

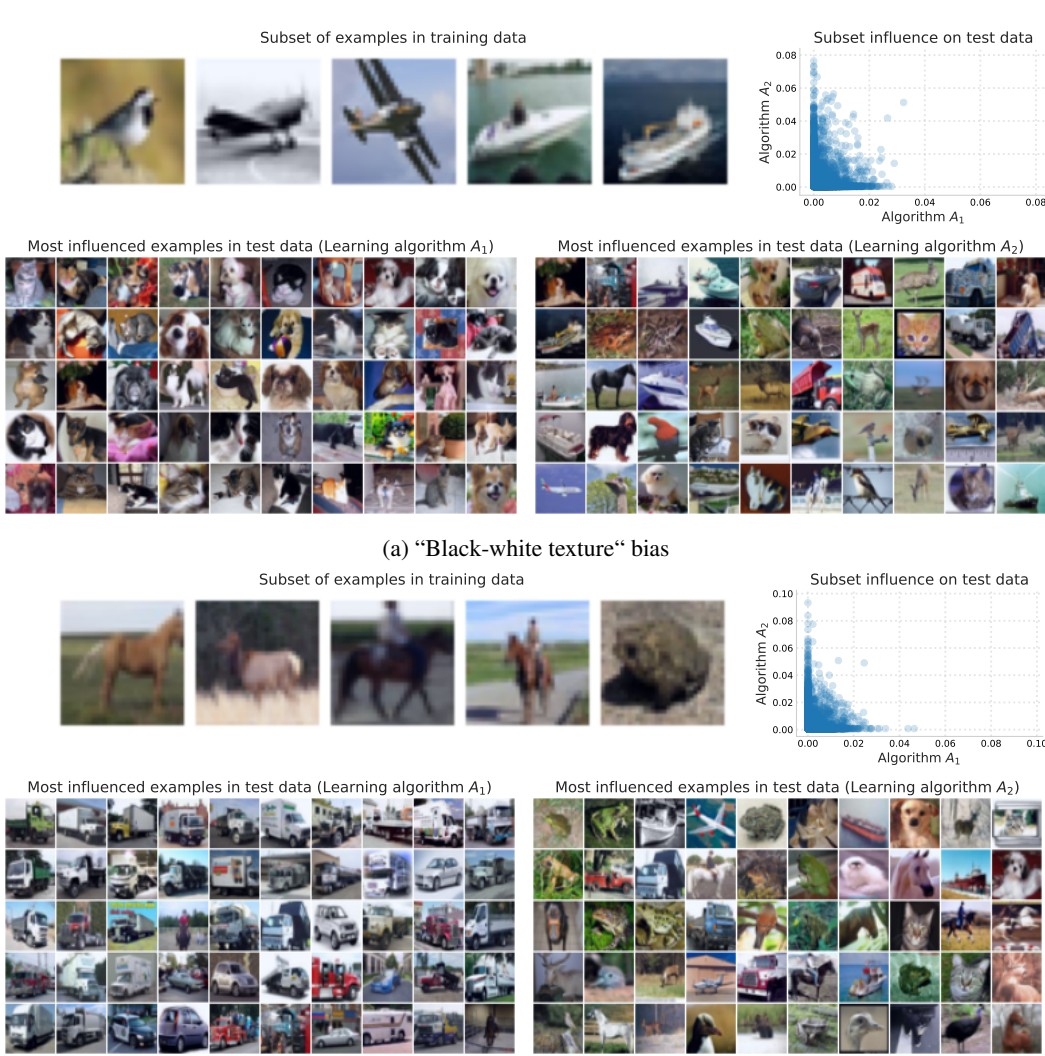

Figure 11: **Relative influence of training data on CIFAR-10 subpopulations**. **Panel (a)**: Training images with black-white objects have high relative influence on the "black-white" dog subpopulation (see Figure 7). These images influence model predictions on test images of black-white dogs (in bottom row) only when models are trained with low SGD noise (alg. $\mathcal{A}_2$). **Panel (b)**: Training images with high-contrast rectangular components in the background have high relative influence on the "rectangular shape" truck subpopulation (see Figure 7). These images influence model predictions on test images of front-facing trucks with prominent rectangular components (in bottom row) only when models are trained with low SGD noise (alg. $\mathcal{A}_2$).

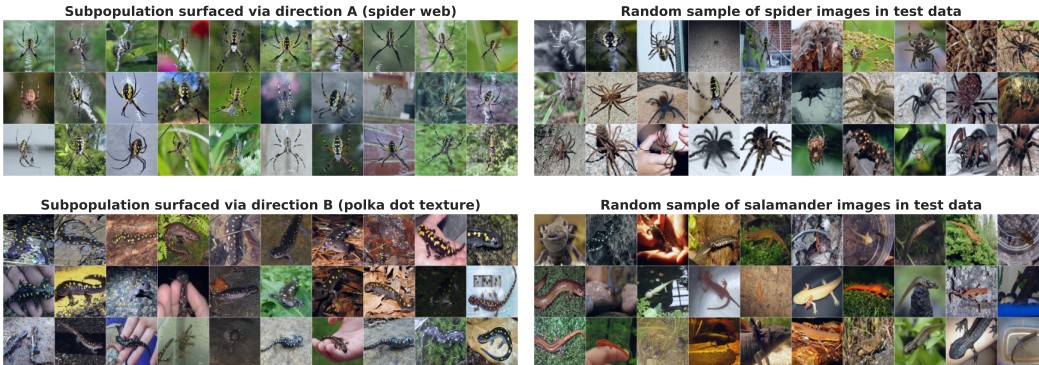

Figure 12: Class-specific visual inspection of LIVING17 subpopulations. **(Top)** In contrast to random LIVING17 images of spiders, the "spider web" subpopulation surfaces spiders with a prominent spider web in the background. **(Bottom)** Unlike random LIVING17 images of salamanders, the "polka dots" subpopulation surfaces salamanders that have a yellow-black polka dot texture.

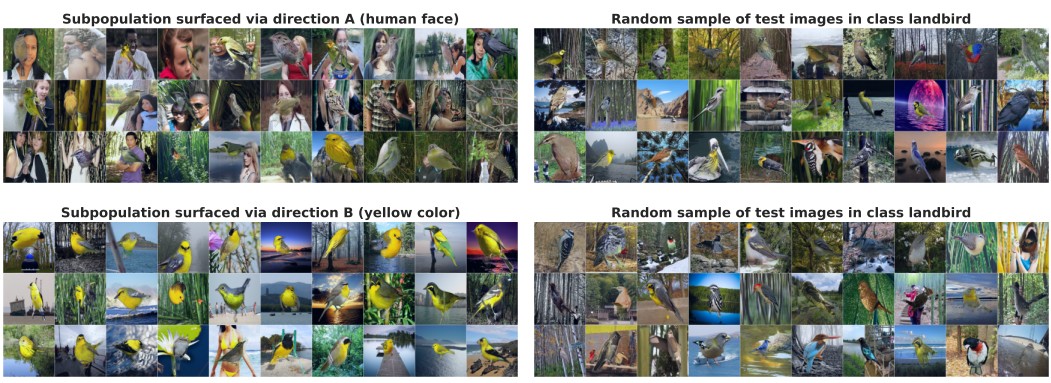

Figure 13: Class-specific visual inspection of WATERBIRDS subpopulations. **(Top)** In contrast to random "landbird" images, the "human face" subpopulation surfaces landbirds with human face(s) in the background. **(Bottom)** Unlike random "landbird" images, the "yellow color" subpopulation surfaces images with yellow birds *or* yellow objects in the background.

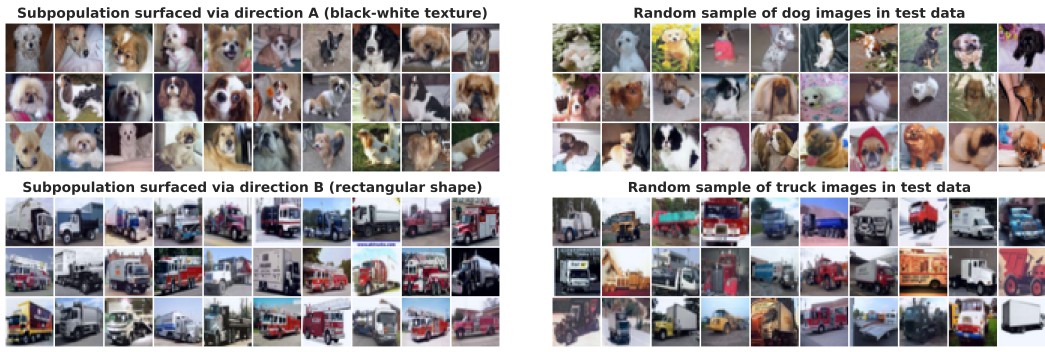

Figure 14: Class-specific visual inspection of CIFAR-10 subpopulations. In contrast to random images of dogs (top) and truck (bottom), the "black-white" and "rectangular shape" subpopulations surface images of black-white dogs and front-facing trucks with multiple rectangular components respectively.

# E   ADDITIONAL EVALUATION OF DISTINGUISHING FEATURE TRANSFORMATIONS

Distinguishing feature transformations, which we recall from Section 2, are functions that, when applied to data points, change the predictions of one model class—but not the other—in a consistent way. In our case studies, we design distinguishing feature transformations that counterfactually verify features that are identified via human-in-the-loop analysis. Our findings in Section 3 use feature transformations to quantitatively measure the relative effect of the identified features on models trained with different learning algorithms. In this section, we present additional findings on feature transformations for each case study:

## E.1   CASE STUDY: STANDARD DATA AUGMENTATION

In Section 3.1, we showed that standard data augmentation—horizontal flips and random crops— amplifies LIVING17 models' reliance on "spider web" and "polka dots" to predict spiders and salamanders respectively. Figure 15 verifies our findings over a larger range of perturbation intensity $\delta$ values. We also observe that decreasing the minimum allowable crop size in `RandomResizedCrop` from 1.0 (i.e., no random cropping) to 0.08 (default `torchvision` hyperparameter) increases models' sensitivity to both feature transformations.

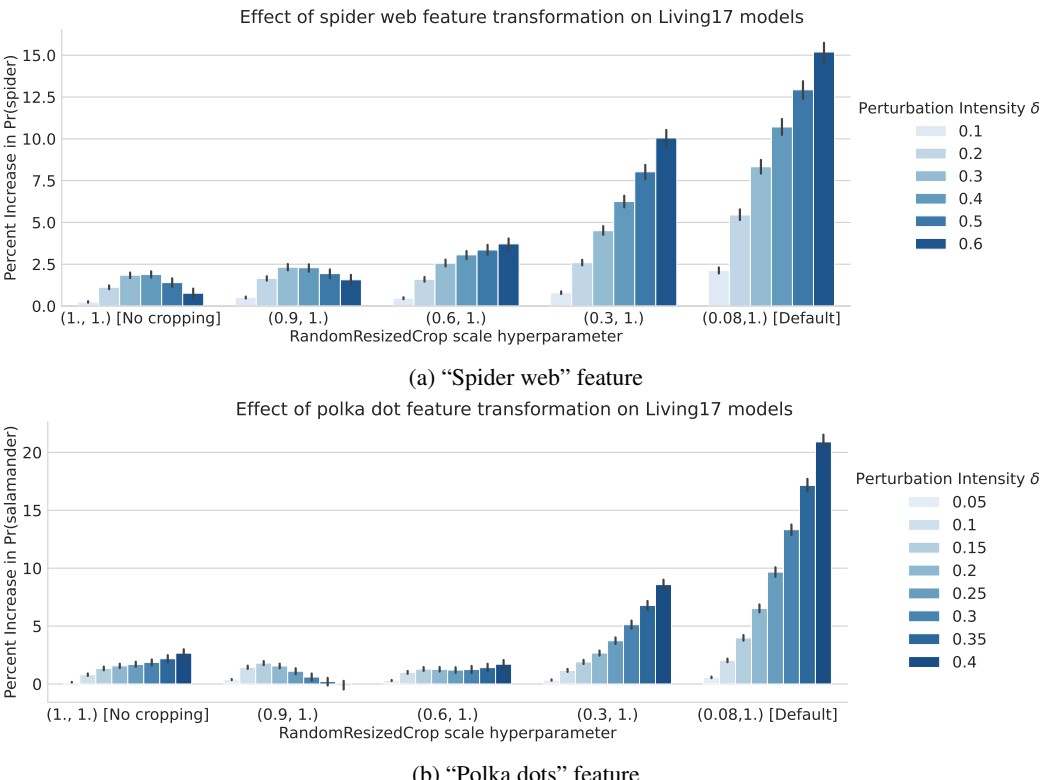

(a) "Spider web" feature

(b) "Polka dots" feature

Figure 15: **Additional evaluation of LIVING17 feature transformations.** The top and bottom row evaluate the effect of "spider web" and "polka dot" feature transformations on models trained with different data augmentation schemes. Increasing the intensity of the transformations and the minimum crop size of `RandomResizedCrop` augmentation (via `scale` hyperparameter) increases the sensitivity of models to both feature transformations in a consistent manner.

### E.2 CASE STUDY: IMAGENET PRE-TRAINING

In Section 3.2, we showed that fine-tuning ImageNet-pretrained ResNet50 models on WATERBIRDS data instead of training from scratch alters the relative importance of two spurious features: "yellow color" and "human face". In Figure 16, we show that both feature transformations alter the predictions of ImageNet-pretrained ResNet18 and ImageNet-pretrained ResNet50 models in a similar way.

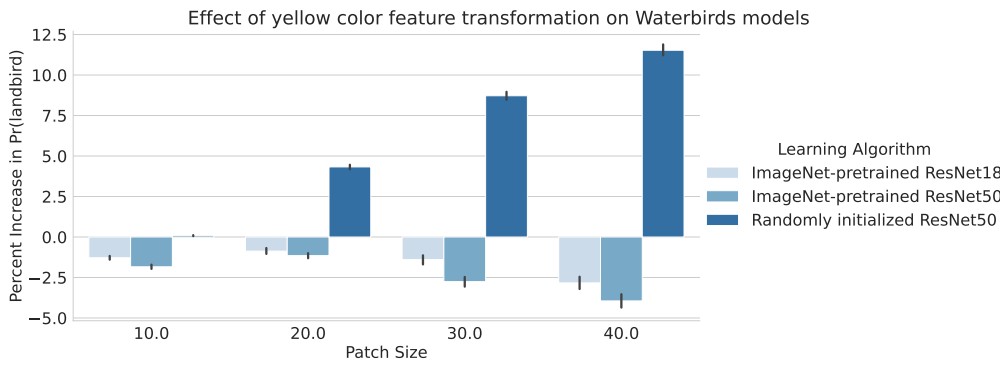

(a) "Yellow color" feature

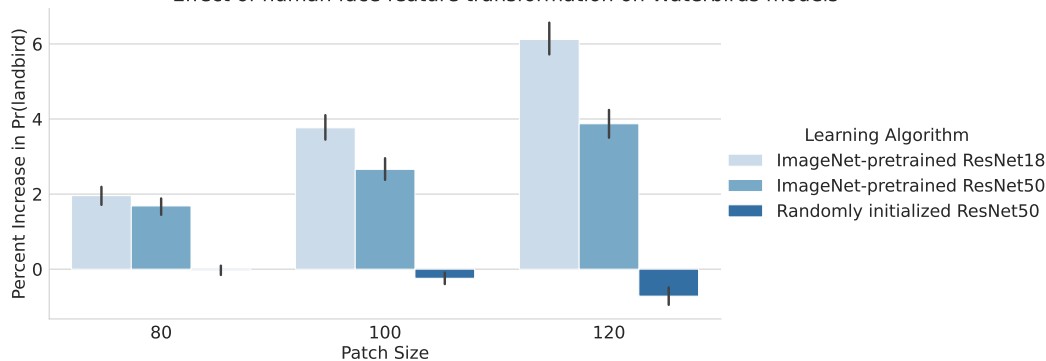

(b) "Human face" feature

Figure 16: **Additional evaluation of WATERBIRDS feature transformations.** The top and bottom row evaluate the effect of "yellow color" and "human face" feature transformations on models trained with and without ImageNet pre-training. In both cases, unlike ResNet50 models trained from scratch, ImageNet-pretrained ResNet18 and ResNet50 models are sensitive to the "human face" transformation but not to the "yellow color" transformation.

### E.3 CASE STUDY: SGD HYPERPARAMETERS

In Section 3.3, we showed that reducing SGD noise results in CIFAR-10 models that are more sensitive to certain features, such as rectangular shape bias and black-white texture to predict trucks and dogs. In Figure 17, we evaluate how feature transformations change class-wise predictions of models trained with different SGD learning rate and batch size hyperparameters.

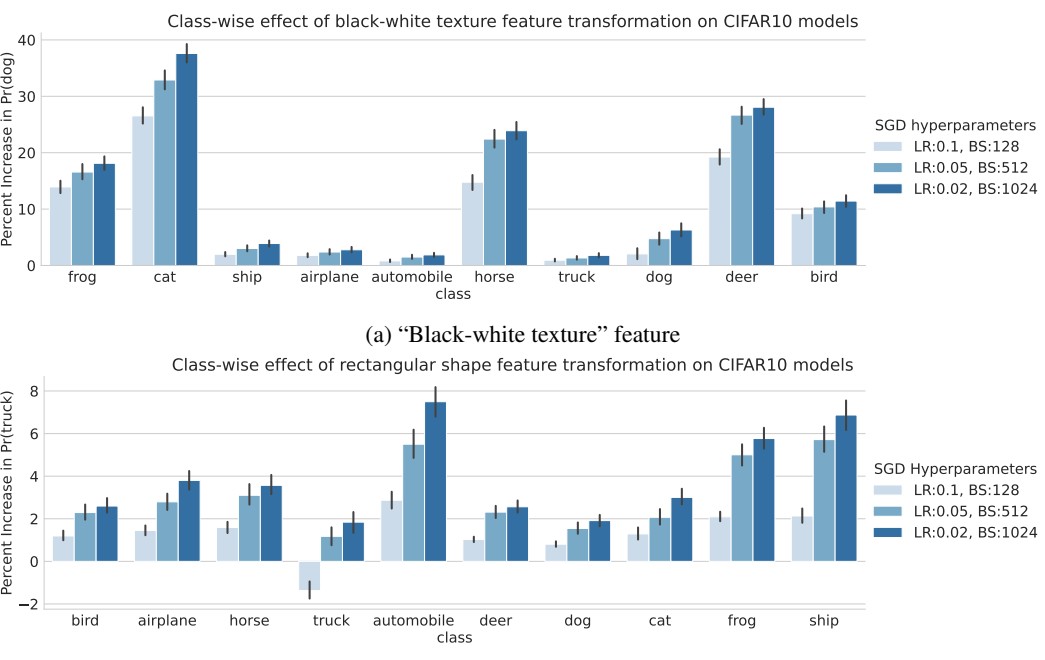

(a) "Black-white texture" feature

(b) "Rectangular shape" feature

Figure 17: **Additional evaluation of CIFAR-10 feature transformations.** The top and bottom row evaluate the effect of "black-white texture" and "rectangular shape" feature transformations on CIFAR-10 models trained with high (light blue), medium, and low (dark blue) SGD noise. In both cases, models trained with higher SGD noise are, on average, more sensitive to these transformations across all classes. Furthermore, the effect of the transformations are class-dependent—model predictions on transformed examples from semantically similar classes differ to a greater extent.

# F MISCELLANEOUS RESULTS

## F.1 AGGREGATE METRIC FOR ALGORITHM COMPARISON

As discussed in Section 4, we can repurpose our framework as a similarity metric that quantifies the similarity of models trained with different learning algorithms in a more global manner. A straightforward approach to output a similarity score (or distribution) is to compute the cosine similarity of datamodel vectors. More concretely, let $\theta_i^{(1)}$ and $\theta_i^{(2)}$ denote the datamodels of example $x_i$ with respect to models trained using learning algorithms $\mathcal{A}_1$ and $\mathcal{A}_2$. Then, the cosine similarity between $\theta_i^{(1)}$ and $\theta_i^{(2)}$ measures the extent to which models trained with $\mathcal{A}_1$ and $\mathcal{A}_2$ depend on the same set of training examples to make predictions on example $x_i$.

We apply this metric to two case studies—pre-training (WATERBIRDS) and SGD noise (CIFAR-10)—in Figure 18. Specifically, Figure 18 plots the distribution of cosine similarity of datamodels for multiple learning algorithms and over all test examples. The left subplot shows that on WATERBIRDS, ImageNet-pretrained ResNet50 models are, on average, more similar to ImageNet-pretrained ResNet18 models than to ResNet50 models pretrained on synthetically generated data (Baradad Jurjo et al., 2021) and models trained from scratch. The right subplot shows that on CIFAR-10, ResNet9 models trained with high SGD noise are more similar to smaller-width ResNet9 models trained with high SGD noise than to ResNet9 models trained with low SGD noise.

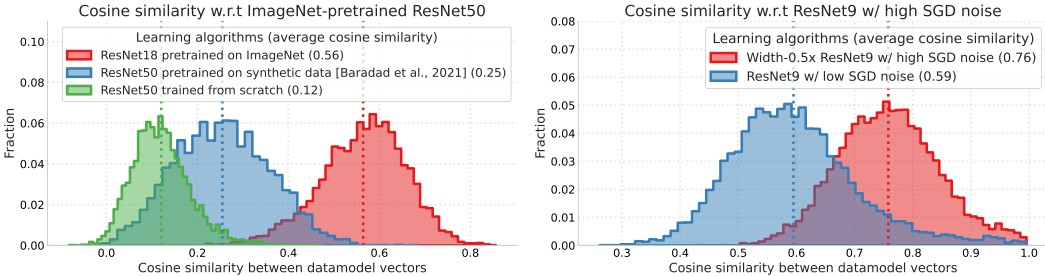

Figure 18: **Datamodel cosine similarity.** We use cosine similarity between two datamodel vectors as an aggregate metric to quantify the similarity of models trained with different learning algorithms. **(Left)** On WATERBIRDS data, datamodels of ImageNet-pretrained ResNet50 and ResNet18 models are more similar to each other than to models pretrained on synthetically generated data and models trained from scratch. **(Right)** On CIFAR-10 data, ResNet models trained with high SGD noise are more similar to each other to ResNet models trained with low SGD noise.

## F.2 EXPLAINED VARIANCE OF RESIDUAL DATAMODEL PRINCIPAL COMPONENTS

Recall from Appendix A that the fraction of variance in datamodel representations $\{\theta_x^{(i)}\}$ explained by training direction $v$ signifies the importance of the direction (or, combination of training examples) to predictions of models trained with algorithm $\mathcal{A}_i$. Through our case studies in Section 3, we show that the top $5-6$ principal components (PCs) of residual datamodels $\theta^{(1\backslash 2)}$ correspond to training directions that have high explained w.r.t. datamodels of algorithm $\mathcal{A}_1$ but not $\mathcal{A}_2$, and vice versa. Figure 19 shows that the top-100 PCs of residual datamodel $\theta^{(1\backslash 2)}$ (resp., $\theta^{(2\backslash 1)}$) have more (resp., less) explained variance on datamodel $\theta^{(1)}$ than on datamodel $\theta^{(2)}$.

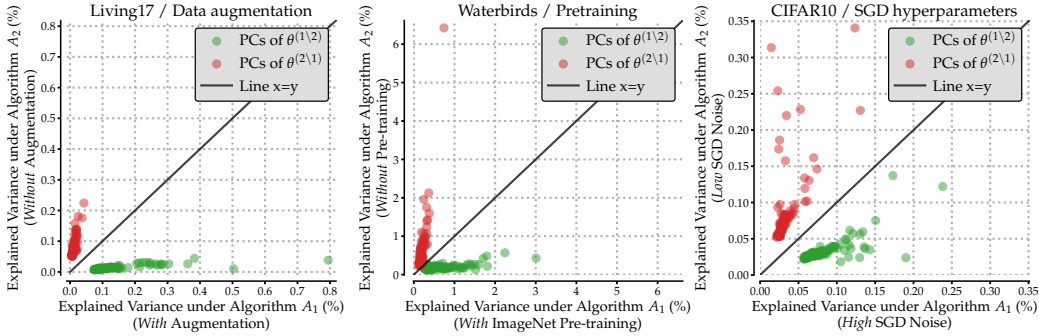

Figure 19: **Explained variance of residual datamodels' principal components.** Highlighted in green (resp. red), the top-100 PCs of residual datamodel $\theta^{(1\backslash 2)}$ (resp. $\theta^{(2\backslash 1)}$) explain a larger (resp. smaller) fraction of datamodel variance under algorithm $\mathcal{A}_1$ than under algorithm $\mathcal{A}_2$ across all three case studies.

F.3    SUBPOPULATIONS SURFACED BY PRINCIPAL COMPONENTS OF RESIDUAL DATAMODELS

As outlined in Section 2, the human-in-the-loop stage of our framework involves extracting test data subpopulations from principal components (PCs) of residual datamodels. Specifically, these subpopulations correspond to test examples whose residual datamodel representations have the most positive (top-$k$) and most negative (bottom-$k$) projection onto a given PC. Here, we show that the top-$k$ and bottom-$k$ subpopulations corresponding to the top few PCs of residual datamodels considered in Section 3 surface test examples with qualitatively similar properties.

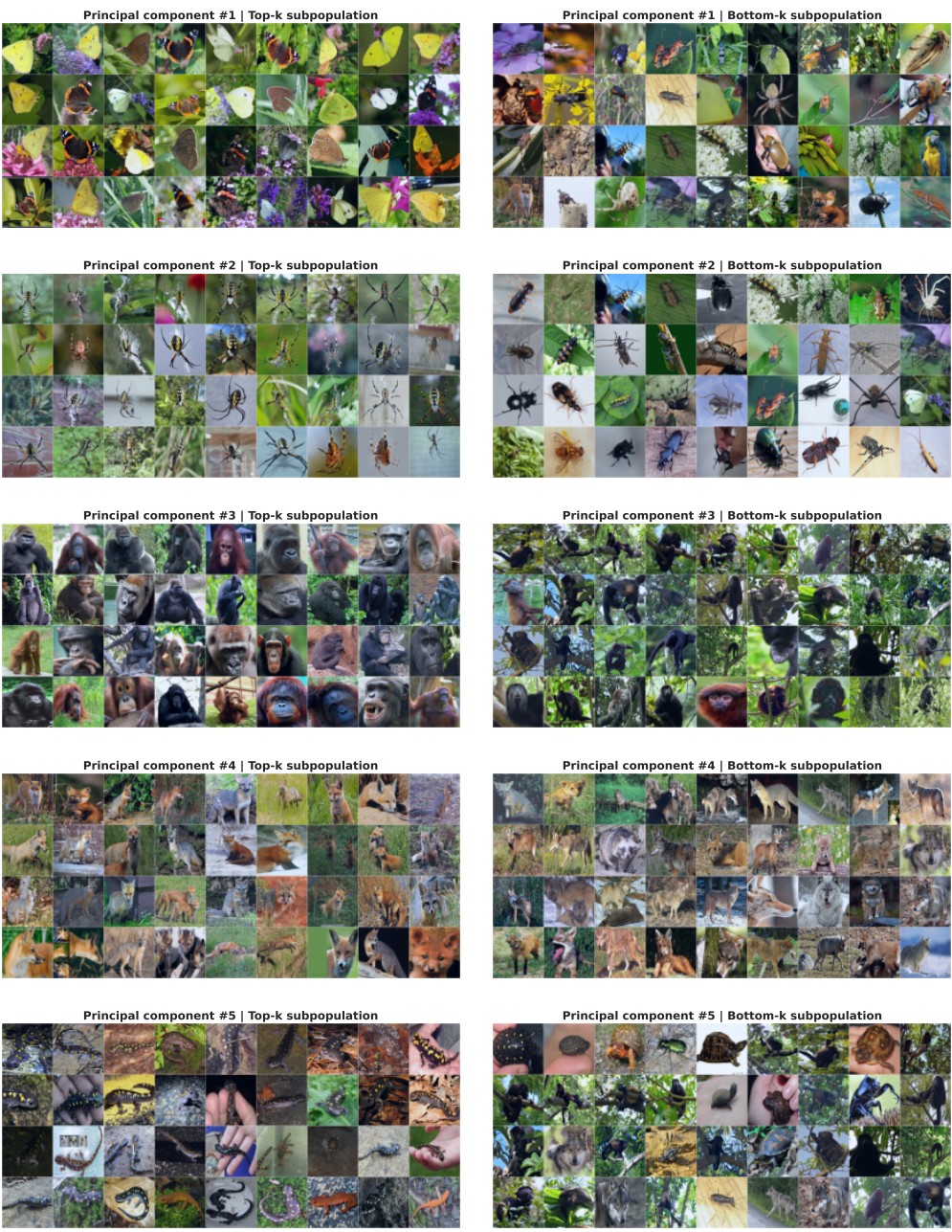

Figure 20: Top five PC subpopulations of LIVING17 residual datamodel $\theta^{(1\backslash 2)}$, where learning algorithms $\mathcal{A}_1$ and $\mathcal{A}_2$ correspond to training models with and without standard data augmentation respectively. Our case study in Section 3.1 analyzes PC #2 (direction **A**) and PC #5 (direction **B**).

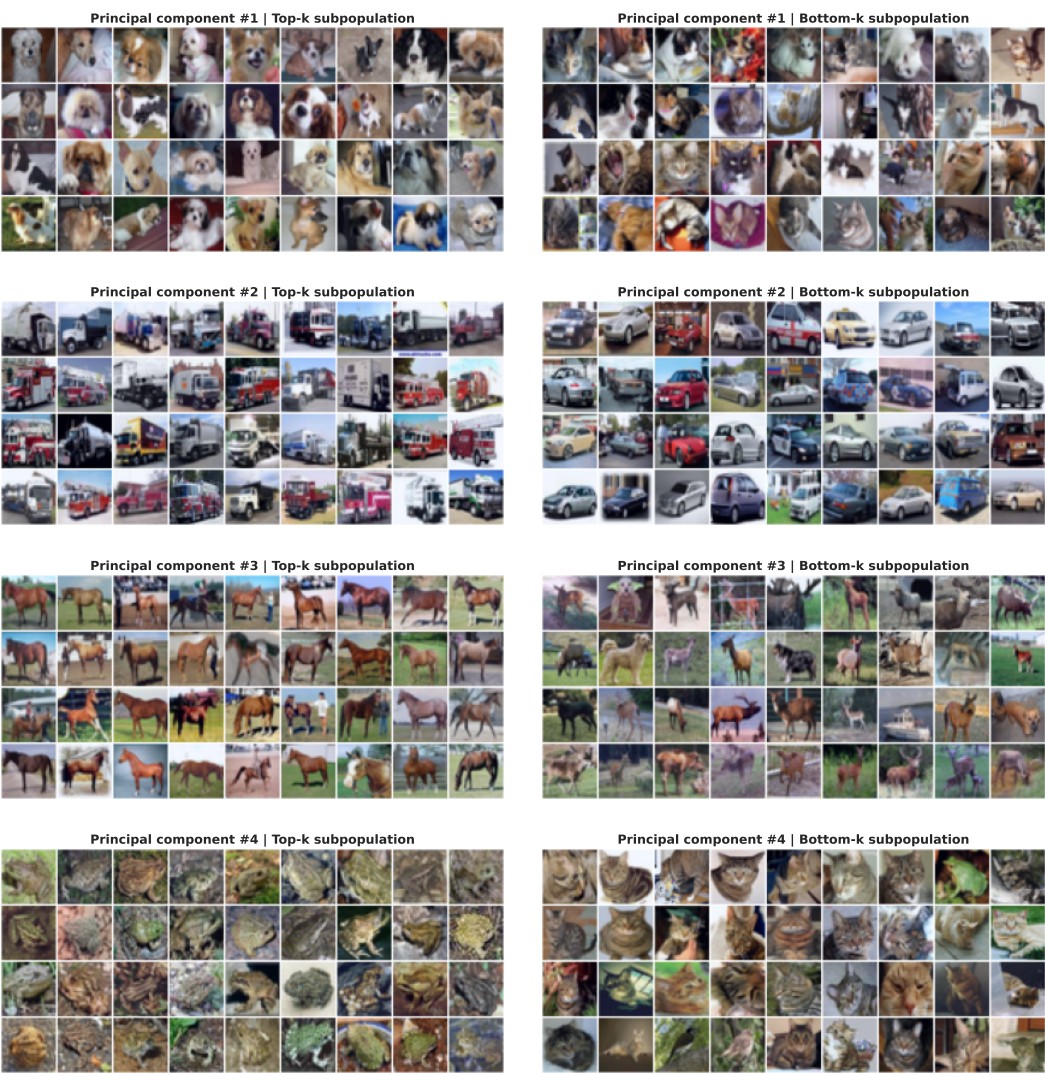

Figure 21: Top four PC subpopulations of CIFAR-10 residual datamodel $\theta^{(2\setminus 1)}$, where learning algorithms $\mathcal{A}_1$ and $\mathcal{A}_2$ correspond to training models with high and low SGD noise respectively. Our case study in Section 3.3 analyzes PC #1 (direction **A**) and PC #2 (direction **B**).

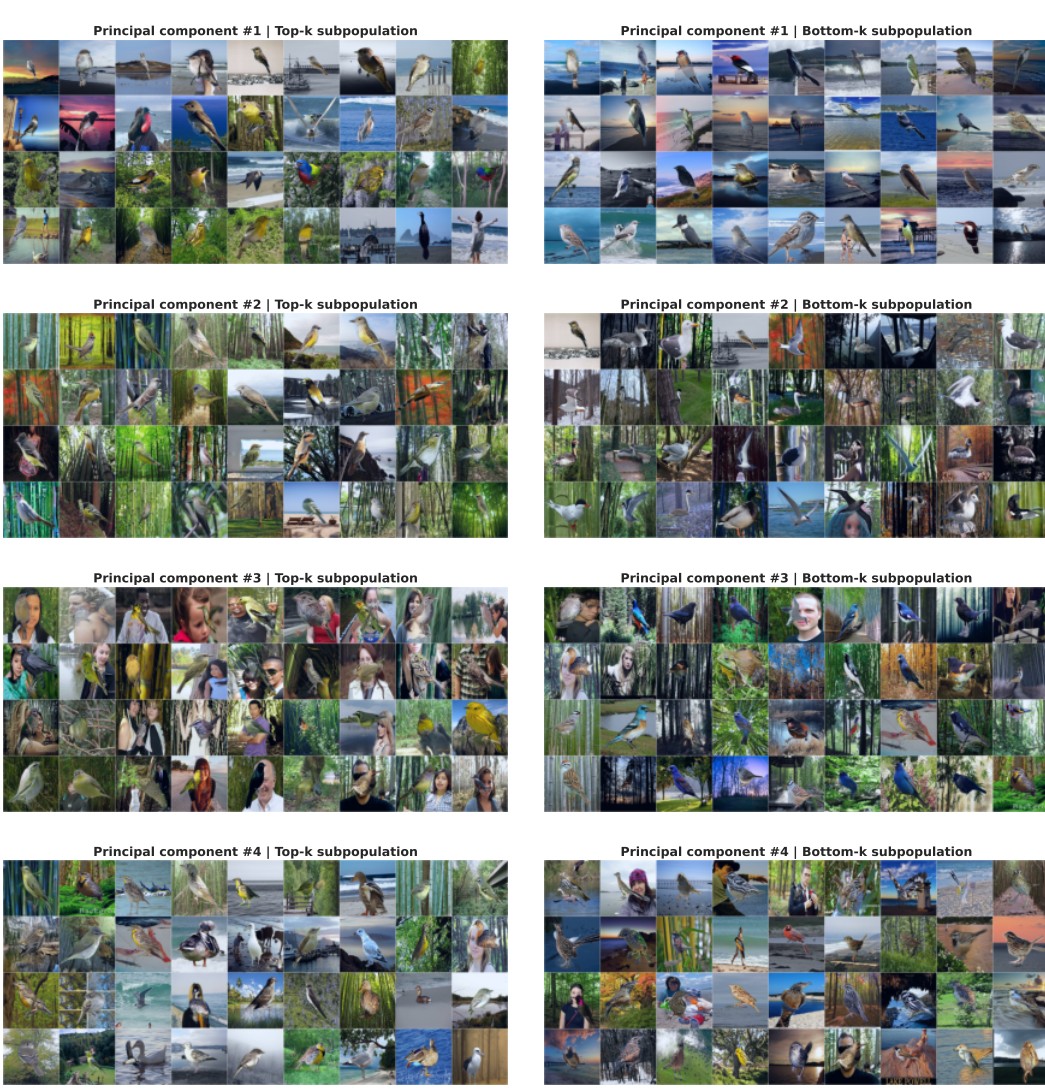

Figure 22: Top four PC subpopulations of WATERBIRDS residual datamodel $\theta^{(1\setminus 2)}$, where learning algorithms $\mathcal{A}_1$ and $\mathcal{A}_2$ correspond to training models with and without ImageNet pre-training respectively. Our case study in Section 3.1 analyzes PC #3 (direction **A**).

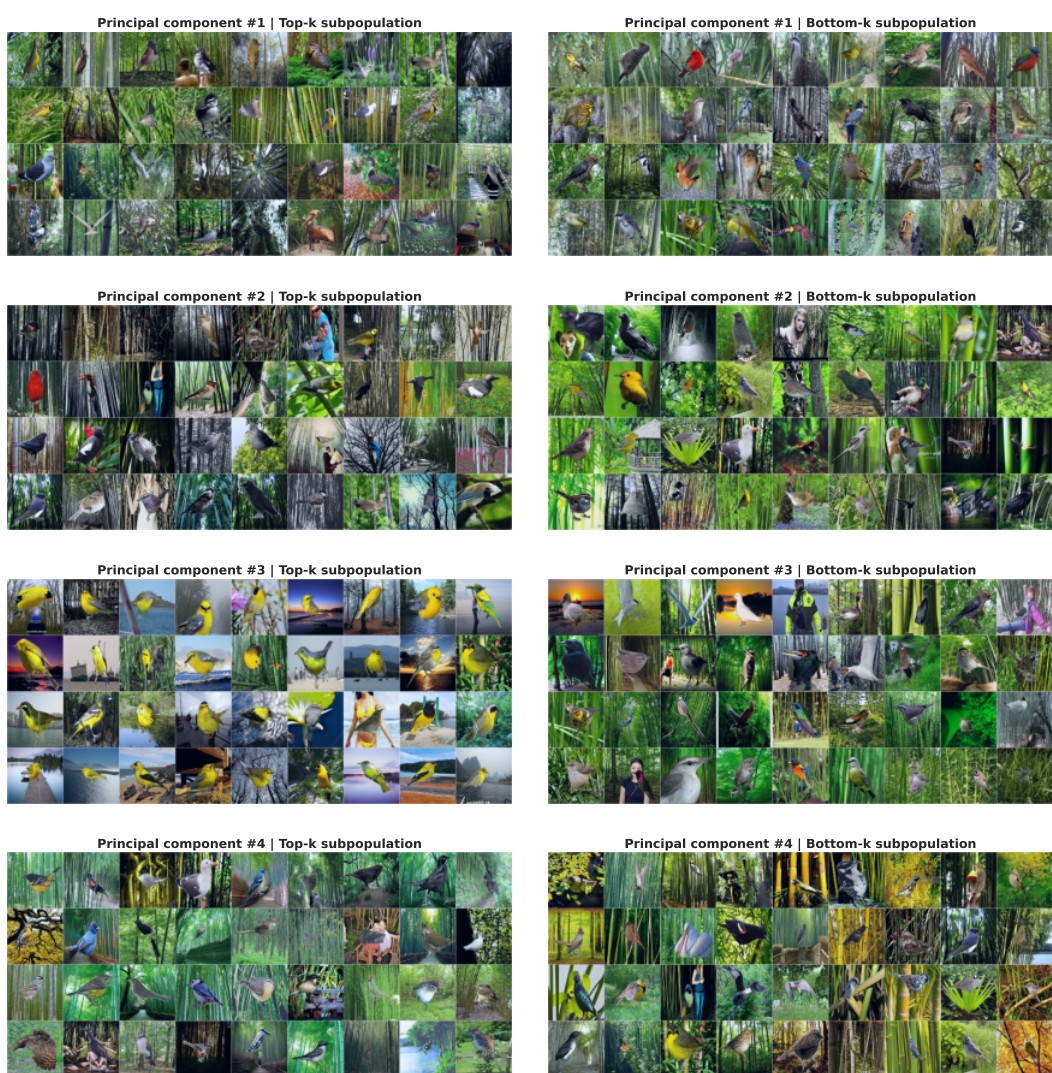

Figure 23: Top four PC subpopulations of WATERBIRDS residual datamodel $\theta^{(2\setminus1)}$, where learning algorithms $\mathcal{A}_1$ and $\mathcal{A}_2$ correspond to training models with and without ImageNet pre-training respectively. Our case study in Section 3.1 analyzes PC #3 (direction **B**).

### F.4 USING CLIP TO IDENTIFY DISTINGUISHING FEATURE CANDIDATES

As discussed in Section 2 and Appendix D, the second stage of our framework applies human-in-the-loop analysis to infer distinguishing feature transformations that disparately impact predictions of models trained with different learning algorithms. In this section, we demonstrate that for image classifiers, shared vision-language models such as CLIP (Radford et al., 2021) provide a streamlined alternative to manual human-in-the-loop identification of distinguishing features.

**Approach**. Before we describe our approach, note that CLIP is a contrastive learning method that embeds text and natural language into a shared embedding space. Our approach leverages CLIP embeddings to identify multiple *distinguishing captions*—representative descriptions that best contrast a given subpopulation of images from a set of images sampled from the same distribution. In the context of our framework, our CLIP-based approach takes as inputs a distinguishing training direction $v$, a set of images $\mathcal{D}$, and a set of captions $\mathcal{S}$[11], and outputs a set of distinguishing captions $\mathcal{S}' \in \mathcal{S}$ in four steps:

- *Pre-compute image and text embeddings*. Use the image encoder of a CLIP model to compute a set of normalized embeddings for all images in $\mathcal{D}$. Analogously, use the text encoder of a CLIP model to compute a set of normalized embeddings for all captions in $\mathcal{S}$.

- *Record image-text pairwise cosine similarity*. Let vector $C_i \in \mathcal{R}^{|\mathcal{S}|}$ denote the pairwise cosine similarity between the embedding of image $i \in \mathcal{D}$ and all captions $j \in \mathcal{S}$.

- *Compute mean cosine similarity over dataset and top-$k$ subpopulation*. Compute the mean cosine similarity vector $\bar{C} = \frac{1}{n} \sum_{i \in \mathcal{D}} C_i$ over all images in $\mathcal{D}$. Similarly, given distinguishing training direction $v$, compute the mean cosine similarity vector $C^{(v)}$ over the top-$k$ images whose residual datamodel vectors are most aligned with $v$.

- *Extract distinguishing captions $\mathcal{S}'$*. Use cosine similarity vectors $\bar{C}$ and $C^{(v)}$ to extract captions in $\mathcal{S}$ that have the maximum difference between $C_i^{(v)}$ and $\bar{C}_i$.

Intuitively, the set of distinguishing captions $\mathcal{S}'$ correspond to representative captions (or, descriptions) that best contrast the top-$k$ images surfaced by distinguishing direction $v$ from the dataset.

**Results**. We now apply this approach to our case study on ImageNet pre-training, where we compare WATERBIRDS models trained with and without ImageNet pre-training (see Section 3). Specifically, we evaluate whether the CLIP-based approach surfaces distinguishing captions that are similar to distinguishing features "yellow color" (direction **A**) and "human face" (direction **B**) inferred via manual human-in-the-loop analysis. Figure 24 illustrates that for direction **A**), the CLIP-based approach highlights distinguishing captions such as `yellow`, `lemon`, and `sulphur`, all of which are similar to the "yellow color" feature that we infer via human-in-the-loop analysis. Similarly, Figure 25 shows that the distinguishing captions for direction **B** (e.g., `florist`, `faces`, `counselors`) are similar to the identified "human face" feature.

To summarize, we demonstrate how the second stage of our framework can be easily *specialized* to comparisons of vision classifiers trained on ImageNet-like data via vision-language embeddings such as CLIP.

---

[11]We use a filtered list of roughly 20,000 most common English words in order of frequency, taken from https://github.com/first20hours/google-10000-english.

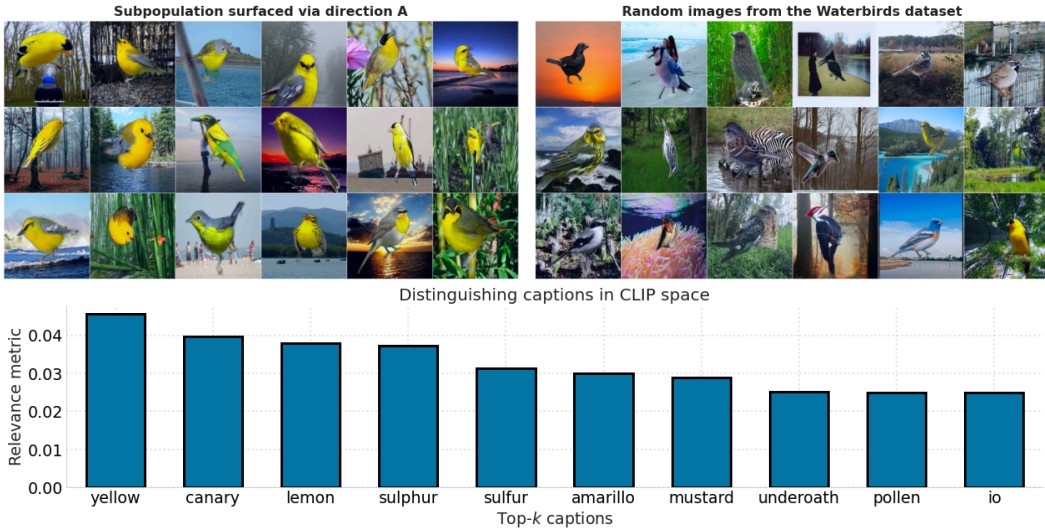

Figure 24: **Direction A**. The CLIP-based approach extracts distinguishing captions such as `yellow`, `lemon`, and `sulphur`, all of which contrast the residual subpopulation on the left to a set of random images from the WATERBIRDS dataset on the right. These distinguishing captions match the "yellow color" feature that we infer and counterfactually verify via human-in-the-loop analysis in Section 3.2.

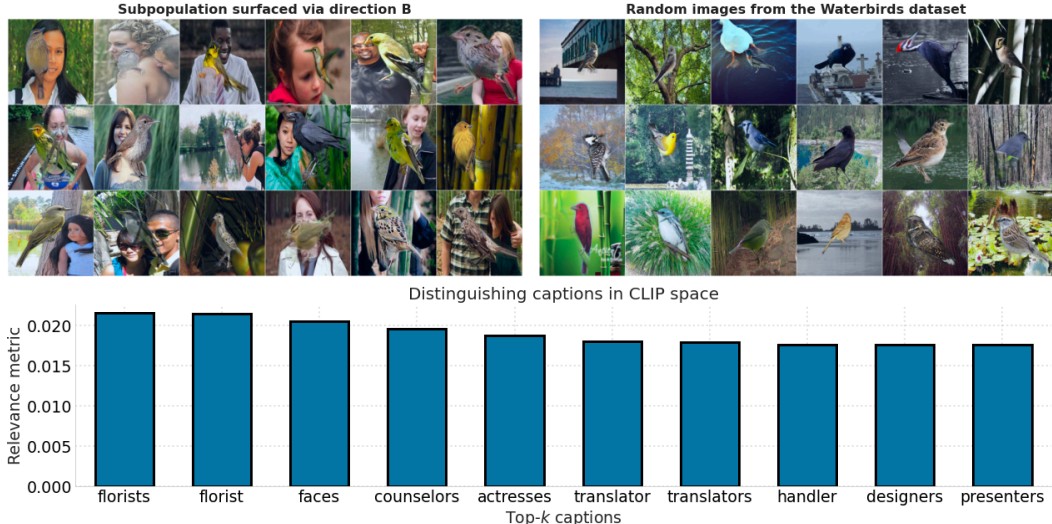

Figure 25: **Direction B**. The CLIP-based approach extracts distinguishing captions such as `florists`, `faces`, and `counselors`, all of which contrast the residual subpopulation (left) of images with human faces in the background to a set of random images (right) from the WA-TERBIRDS dataset. These distinguishing captions match the "human face" feature that we infer and counterfactually verify via human-in-the-loop analysis in Section 3.2.

### F.5 ADDITIONAL ANALYSIS ON THE EFFECT OF IMAGENET PRE-TRAINING

In this section, we evaluate our algorithm comparisons framework with a controlled experiment on WATERBIRDS data, wherein we compare learning algorithms that differ in one or more axes. Through this experiment, we show that our framework identifies feature transformations that (a) impact similar algorithms similarly and (b) distinguish dissimilar algorithms even when they result in models with similar (and relatively high) test accuracies.

**Setup**. To design a controlled experiment, we extend our case study on ImageNet pre-training (see Section 3.2) in two ways:

- **Learning algorithms**. We consider four learning algorithms in this experiment. Similar to our case study in Section 3.2, algorithms $\mathcal{A}_1$ and $\mathcal{A}_2$ correspond to ResNet50 models pre-trained on ImageNet and ResNet50 models trained from scratch. The two additional algorithms control for (a) choice of pre-training data and (b) usage of pre-training. In particular, algorithm $\mathcal{A}_3$ corresponds to ResNet18 models pre-trained on ImageNet and algorithm $\mathcal{A}_4$ corresponds to ResNet50 models pre-trained on synthetically generated data (Baradad Jurjo et al., 2021).
- **Dataset**. We restrict our analysis to the majority group (i.e., landbirds on land) of the WATER-BIRDS dataset. With this modification, we control for aggregate performance—models trained with any of the four algorithms attain at least 97% accuracy on this group.

In other words, the learning algorithms described above output models with similar aggregate performance but differ in multiple axes; $\mathcal{A}_1$ and $\mathcal{A}_3$ pre-train on the same dataset (ImageNet), but differ in model architecture; algorithms $\mathcal{A}_1$ and $\mathcal{A}_4$ pre-train but on different datasets; algorithms $\mathcal{A}_1$ and $\mathcal{A}_2$ differ in their usage of pre-training.

**Explained variance**. Recall that *fraction of explained variance* of a given vector $v \in \mathbb{R}^d$ in a set of vectors $\{\theta_i \in \mathbb{R}^d\}$ is the empirical variance of $v^\top \theta_i$ divided by the total amount of variance in $\{\theta_i\}$ (i.e., trace($\text{Cov}[\theta_i]$)). In other words, this measures what fraction of the total variation in $\{\theta_i\}$ is along the direction $v$.

**Comparing direction-specific explained variance**. Recall that through our case study on ImageNet pre-training in Section 3.2, we identified (and counterfactually verified) a "human face" distinguishing feature that impact models pre-trained on ImageNet (algorithm $\mathcal{A}_1$) but not models trained from scratch (algorithm $\mathcal{A}_2$). Here, we consider the fraction of datamodel variance explained by the training direction corresponding to the "human face" feature across all four learning algorithms. Figure 26 shows that the fraction of datamodel variance explained by this direction is maximum under algorithms $\mathcal{A}_1$, followed by algorithm $\mathcal{A}_3$—both algorithms that pre-train ResNet models on ImageNet.

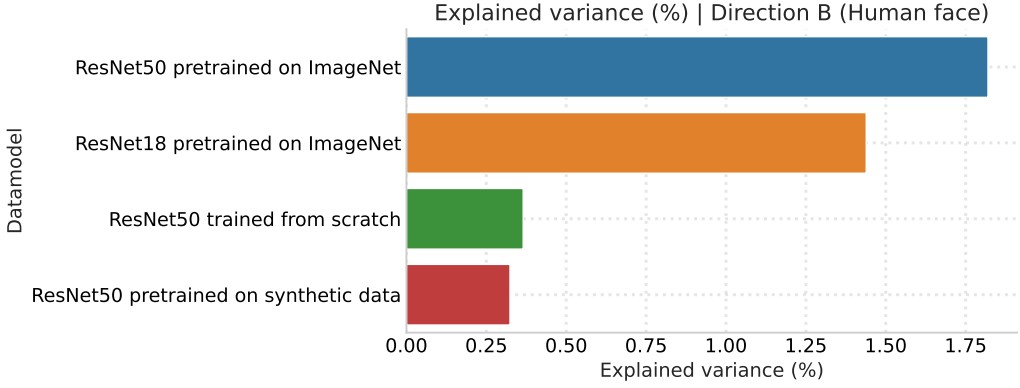

Figure 26: Fraction of datamodel variance explained by the training direction corresponding to the "human face" feature in Section 3.2. Each set of datamodel representations maps to one of four different learning algorithms, which differ in terms of architecture, pre-training, and/or choice of pre-training dataset.

**Comparing top-$k$ explained variance**. For a given set of training directions $\{v_1, ..., v_k\}$, the fraction of explained variance in datamodel representations captures the importance of the training directions to model predictions for algorithm $\mathcal{A}_i$. In Figure 27, we compute the top-$k$ principal components of datamodels corresponding to algorithms that pre-train on ImageNet—$\mathcal{A}_1$ and $\mathcal{A}_3$—and evaluate the fraction of explained variance in datamodel representations as a function of the number of directions $k$ and the learning algorithm $\mathcal{A}_i$. As expected, the PC directions have high explained variance on datamodels corresponding to algorithms that pre-train on ImageNet and low explained variance on datamodels corresponding to algorithms that pre-train on synthetic data and train from scratch.

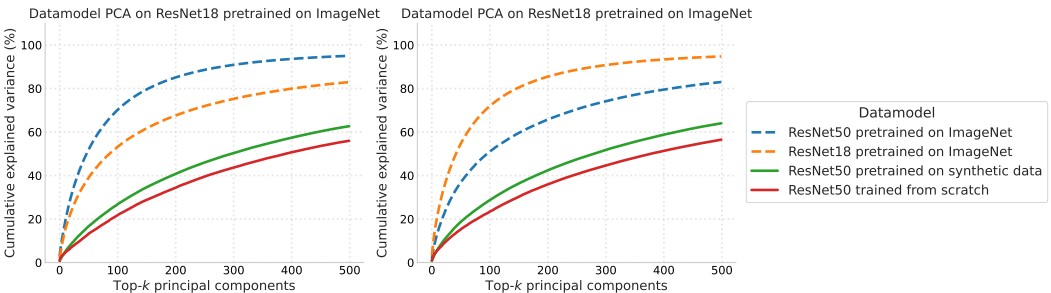

Figure 27: Given number of directions $k$ and the learning algorithm $\mathcal{A}_i$, we compute the top-$k$ principal components of datamodels corresponding to algorithms that pre-train on ImageNet—$\mathcal{A}_1$ (left) and $\mathcal{A}_3$ (right)—and evaluate the fraction of explained variance in datamodel representations corresponding to algorithm $\mathcal{A}_i$. These PCs have high explained variance on datamodels corresponding to algorithms pre-trained on ImageNet and low explained variance on datamodels corresponding to models pre-trained on synthetic data and models trained from scratch.

Note that these results are consistent with our findings on (a) aggregate algorithm comparisons via datamodel cosine similarity in Appendix F.1 and (b) additional counterfactual analysis of distinguishing feature transformations in Appendix C.4.

### F.6 ON COMPARISONS WITH MODEL PREDICTIONS AND PENULTIMATE-LAYER REPRESENTATIONS

In this section, we contrast our algorithm comparisons framework to comparisons based on model predictions and penultimate-layer representations. Note that there are no existing methods that can be directly reused for comparing learning algorithms to the best of our knowledge. Therefore, we design experiments to evaluate whether model predictions and penultimate-layer representations can identify distinguishing training directions surfaced using our framework.

**Model predictions**. Through this experiment, we show that example-level differences in predictions (Zhong et al., 2021; Meding et al., 2022) of models trained with different algorithms are not necessary to identify subpopulations analysed in our case studies. First, we re-run the first stage of our framework only on test examples on which models trained with different algorithm have the same prediction on average. Then, we compare distinguishing training directions (i.e., output of the first stage) before and after controlling for prediction-level agreement. Our results in Table 1 show that for each case study, our framework identifies similar training directions (i.e., high cosine similarity) even after removing test examples on which model predictions differ. This experiment shows that our framework can identify fine-grained differences between learning algorithms that persist even after controlling for prediction-level disagreement across models trained with different algorithms.

| Dataset / Case study | Direction | (Absolute) Cosine Similarity |
| --- | --- | --- |
| Living17 / Data augmentation | **A** (Spider web) | 0.999 |
|  | **B** (Polka dots) | 0.998 |
| Waterbirds / ImageNet pre-training | **A** (Yellow color) | 0.977 |
|  | **B** (Human face) | 0.740 |
| CIFAR-10 / SGD hyperparameters | **A** (Black-white texture) | 0.998 |
|  | **B** (Rectangular shape) | 0.999 |

Table 2: Distinguishing training directions before and after filtering out high-disagreement test examples exhibit high cosine similarity and surface subpopulations of images that share the same distinguishing feature.

**Penultimate-layer representations**. Representation-based comparisons (Raghu et al., 2017; Morcos et al., 2018b; Kornblith et al., 2019a) measure the degree to which different models' representation can be aligned. Unlike datamodel representations, penultimate-layer representations are not aligned—coordinates of penultimate-layer representations do not share a consistent interpretation across different models. So, we first introduce a variant based on penultimate-layer representations that has a consistent basis. Specifically, similar to how the datamodel weight $\theta_{ij}$ denotes the influence of training example $j$ over the prediction on test example $i$, we set $\theta_{ij}^{(r)}$ to equal the cosine similarity between the penultimate-layer representation of test example $i$ and train example $j$. We then compare two properties of datamodel representations and penultimate-layer representations:

- **Effective dimensionality of representations**: We first compare the effective dimensionality (i.e., cumulative fraction of variance explained by top-$k$ components) of datamodel representations $\theta$ and penultimate-layer representations $\theta^{(r)}$. Figure 28 shows that for all datasets and learning algorithms, datamodel representations have significantly higher effective dimensionality than the penultimate-layer alternative. For example, on CIFAR-10 data, more than $99\%$ of the variation in penultimate-layer representations is captured by the first 10 components.

- **Explained variance of distinguishing training directions**: We now re-run the first stage of our framework to compare distinguishing directions obtained via datamodel representations $\theta$ and penultimate-layer representations $\theta^{(r)}$. Here, we evaluate the extent to which these training directions distinguish models trained with different algorithms. Specifically, as shown in Figure 29, we compare the difference in the cumulative fraction of variance explained by the top-$k$ training directions across representations corresponding to algorithms $\mathcal{A}_1$ and $\mathcal{A}_2$ (higher the better). Figure 29 shows that (a) training directions obtained from datamodel representations have significantly higher gap in explained variance across learning algorithms and (b) directions obtained

from penultimate-layer representations can have close to zero or negative gap in explained variance across learning algorithms.

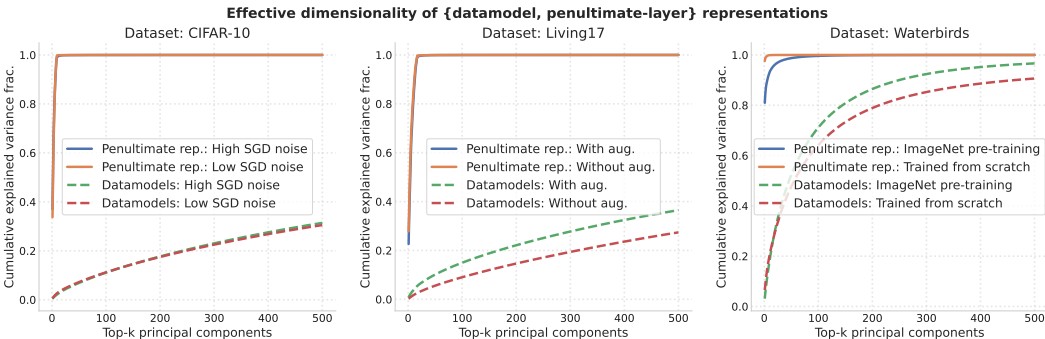

Figure 28: Effective dimensionality (i.e., cumulative fraction of variance explained by top-$k$ components) of datamodel representations is significantly more than that of penultimate-layer representations across all datasets and learning algorithms considered in Section 3.

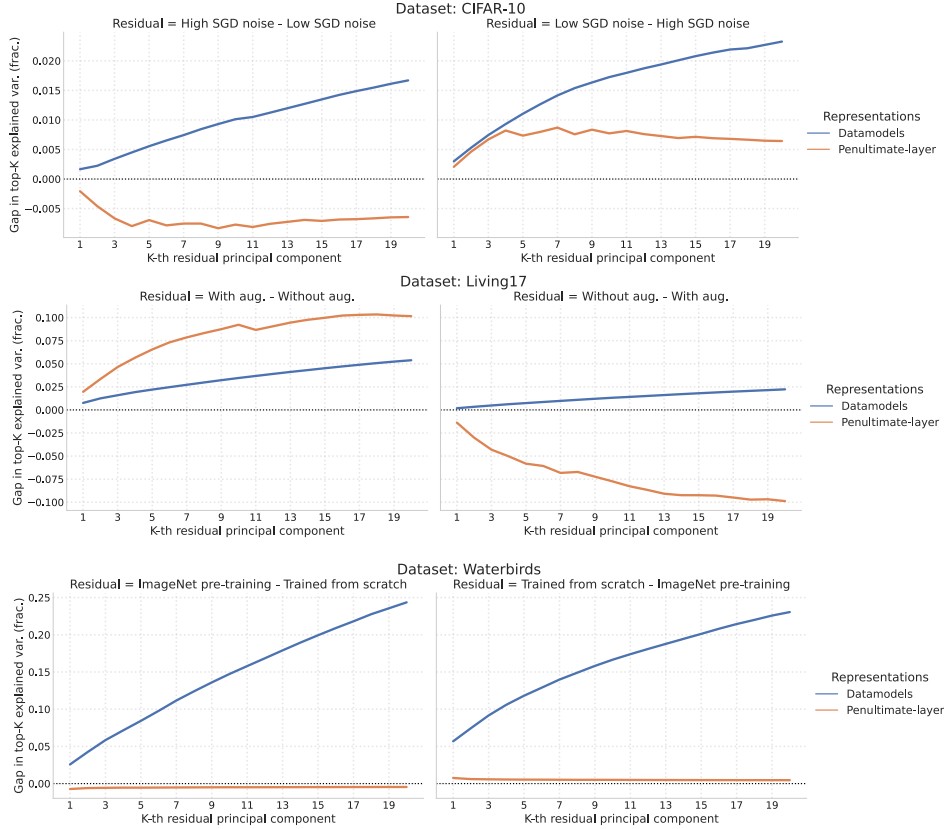

Figure 29: Difference in the cumulative fraction of variance explained by the top-$k$ training directions across (datamodel or penultimate-layer) representations corresponding to learning algorithms $\mathcal{A}_1$ and $\mathcal{A}_2$; higher the better. Top-$k$ distinguishing training directions obtained from datamodel representations have significantly higher gap in explained variance across learning algorithms (e.g., CIFAR-10, WATERBIRDS) and (b) directions obtained from penultimate-layer representations can have close to zero (e.g., WATERBIRDS) or negative gap (e.g., CIFAR-10, LIVING17) in explained variance across learning algorithms.

### F.7 EFFECT OF SAMPLE SIZE ON DATAMODEL ESTIMATION

In this section, we analyze the effect of datamodel sample size on datamodel estimation.

**Setup**. Recall from Appendix C.3 that a datamodel training set of size $m$ corresponds to training $m$ models on independently sampled training data subsets. For our case study on ImageNet pre-training in Section 3.2, we estimate datamodels on WATERBIRDS data with a $50,000$ samples (i.e., $m$ ResNet50 models trained on random subsets of the WATERBIRDS training dataset). In this experiment, we analyze how the estimated datamodels vary as a function of sample size $m \in \{5000, 10000, 25000, 50000\}$.

**Cosine similarity between datamodels**. Our algorithm comparisons framework uses normalized datamodel representations to compute distinguishing training directions in the first stage. So, we first analyze the alignment between datamodel representations that are estimated with different sample sizes. Specifically, we evaluate the cosine similarity between $\theta_x^{(m_1)}$ and $\theta_x^{(m_2)}$, where vector $\theta_x^{(m)} \in \mathbb{R}^{|S|}$ corresponds to the linear datamodel for example $x$ estimated with $m$ samples. As shown in Figure 30, the average cosine similarity between datamodels is greater than $0.9$ even when the sample size is reduced by a factor of $10$, from $50000$ to $5000$.

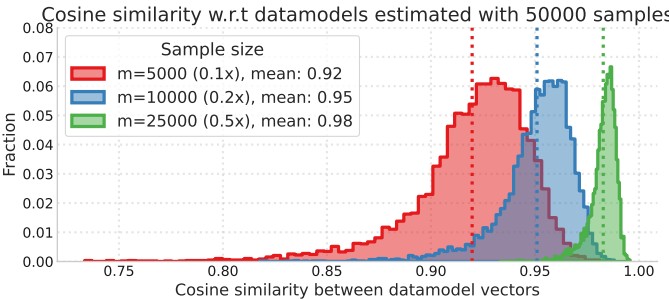

Figure 30: Histogram over cosine similarity between datamodels $\theta_x^{(m_1)}$ and $\theta_x^{(m_2)}$, where vector $\theta_x^{(m)} \in \mathbb{R}^{|S|}$ corresponds to the linear datamodel for example $x$ estimated with $m \in \{5000, 10000, 25000, 5000\}$ samples.

**Explained variance of principal components**. As discussed in Section 3, for a given training direction $v$, the fraction of variance that $v$ explained in datamodel representations $\{\theta_x^{(i)}\}$ captures the importance of the corresponding combination of training examples to model predictions for algorithm $\mathcal{A}_i$. Here, we show that principal components of datamodel representations trained with smaller sample size (e.g., $m = 500$) have similar explained variance on datamodel representations estimated with larger sample size, and vice-versa. As shown in Figure 31, the explained variance of the top-10 principal components of datamodels estimated with $m \in \{500, 5000\}$ have similar explained variance on datamodels estimated with $m \in \{5000, 10000, 25000, 50000\}$.

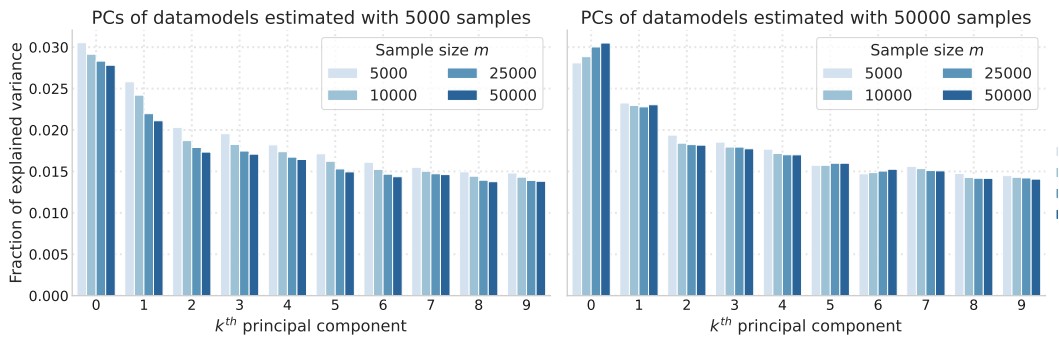

Figure 31: Explained variance of the top-10 principal components of datamodels estimated with $m \in \{500, 5000\}$ have similar explained variance on datamodels estimated with sample size $m \in \{5000, 10000, 25000, 50000\}$.

# G  RELATED WORK ON MODEL INTERPRETATION

In the main paper, we focused our discussion of related work to those directly related to model comparisons. Here, we give discuss additional related work on interpretability and debugging of model biases.

**Interpretability, explainability, and debugging.** Stage II of our method hinges on the interpretability of the extracted subpopulation. A long line of prior work propose different interpretability and explainability methods for models.

Local explanation methods include saliency maps (Simonyan et al., 2013; Dabkowski & Gal, 2017; Adebayo et al., 2018), surrogate models such as LIME (Ribeiro et al., 2016), and Shapley values (Lundberg & Lee, 2017). Our method is similar to per-example based interpretability methods such as influence functions (Koh & Liang, 2017) in that our interpretation is based on data; however, our analysis differs from these priors methods in that it looks at entire subpopulations of inputs.

Global interpretability and debugging methods often leverage the rich latent space of neural networks in order to identify meaningful subpopulations or biases more automatically. Concept activation vectors and its variants (Kim et al., 2018; Abid et al., 2022; Ghorbani et al., 2019) help decompose model predictions into a set of concepts. Other recent works (Eyuboglu et al., 2022; Jain et al., 2022) leverage the recent cross-model representations along with simple models—mixture models and SVMs, respectively—to identify coherent subpopulations or slices. Other methods (Wong et al., 2021; Singla & Feizi, 2021) analyze the neurons of the penultimate layer of (adversarially robust) models to identify spurious features. Our framework can be viewed as leveraging a different embedding space, that of datamodel representations, to analyze model predictions.

**Robustness to specific biases.** In applying our framework across the three case studies, we identify a number of both known and unknown biases. A large body of previous work aims at finding and debugging these biases: Priors works investigate specific biases such as the role of texture (Geirhos et al., 2019) or backgrounds (Xiao et al., 2020) by constructing new datasets. Leclerc et al. (2021) automate many of these studies in the context of vision models with a render-based framework. Crucially, these works rely on having control over data generation and having candidate biases ahead of time. See the "Connections to prior work" within each case study for more specific references.

