# OpenReview forum: "A Unified Framework for Comparing Learning Algorithms"
_ICLR.cc/2023/Conference — Submitted to ICLR 2023_

### Official Review · Reviewer_1DZF · 2022-10-23

**Confidence:** 3
**Correctness:** 3
**Technical Novelty And Significance:** 2
**Empirical Novelty And Significance:** 2
**Recommendation:** 5

**Clarity, Quality, Novelty And Reproducibility:**

- The paper is well-written and easy to follow, but maybe a little wordy, paragraphs before and after section 2.2 title are essentially the same.
- The writing quality is good. The proposed method rather have limited novelty and significance.
- The reproducibility of the human-in-the-loop part of stage II is doutful.


**Strength And Weaknesses:**

Strength
- The algorithm is simple and the paper is overall well-written.
- Comparison among models, instead of studying a single model, is a promising direction for model interpretation.

Weakness
- This work seems like to be related to model interpretation, or explainable machine learning, because it gives a human-understandable interpretation, or explaination of how models predict specific classes. However, related literature reivew and comparions are missing.
- The process seems to heavily rely on the method used in Stage I, 1. Authors also mentioned the influence function, why it is not used but the datamodel is used?
- Stage II seems to be non-deterministic for the human-in-the-loop part. What if different results maybe given by the same output of stage I? This weakens reproducibility somehow.
- Authors lack describing how the method would benefit users in the three application cases. After finding the transformations, how would they give hints on improving any aspects of each scene?

**Summary Of The Paper:**

This paper propose a method to find the differences of how differently trained models make predictions. More precisely, it gives human-interpretable features (or patterns) on how different models recognize different classes. The proposed method heavily relies on how influences of training data on test data are measured. Authors showcase three applications of the proposed method.

**Summary Of The Review:**

Although being well-written, lacking enough discussion on related work and utility of the proposed method, I would like to reject the paper.

---

> ### Author Response · Authors · 2022-11-15
> **Response to Reviewer 1DZF**
>
> We thank the reviewer for their useful comments and suggestions. We address each of the raised weaknesses below:
>
> **“Related to model interpretation…related literature review and comparisons are missing.”** We appreciate the reviewer highlighting the gap in related work on interpretability and model debugging, and we have added more references to the relevant literature (Appendix G). In our initial version, we focused our discussion of related work mostly on those directly related to model comparison, as this is the primary objective of our paper. The initial version also included various references to the specific biases we studied embedded within each section (see the “Connection to previous work” paragraphs in each subsection of Section 3).
>
> **“The process seems to heavily rely on the method used in Stage I…influence function, why [is it] not used?.”** Our comparison method hinges on the unique properties of datamodels. For one, datamodels’ model-agnostic representation allows comparisons across different arbitrary model classes. Furthermore, their ability to approximate model outputs allows us to attribute a precise quantitative interpretation to the result of comparison. We elaborate more precisely on the theoretical properties of datamodels in Appendix A (algorithm analysis).
>
> The reviewer is correct that in principle one could replace datamodel representations with influence estimates (in fact, [IPE+22] show that influences can be viewed as a specific instantiation of datamodel vectors). However, common influence function estimates in deep learning tend to be unreliable or brittle [BPF21], and fail to capture model behavior anywhere near as well as datamodels (see [IPE+22], Figure 7). In particular, there are challenges to making influence functions scale to large-scale settings such as deep neural networks, due to having to approximate the inverse of a Hessian (previous work such as [KL17] circumvent this issue by using only the last layer of the neural network to estimate influence functions). For these reasons, we expect our method to be much less effective if we replace datamodels with (standard) implementations of influence functions.
>
> [KL17] Understanding Black-box Predictions via Influence Functions (https://arxiv.org/abs/1703.04730)
>
> [BPF21] Influence Functions in Deep Learning are Fragile (https://arxiv.org/abs/2006.14651)
>
> [IPE+22] Datamodels: Predicting Predictions from training data (https://arxiv.org/abs/2202.00622)
>
> **“Stage II seems to be non-deterministic for the human-in-the-loop part…this weakens reproducibility:”** We agree with the reviewer that Stage II is not entirely algorithmic and inherently comes with some variability. However, we would like to point out that any analysis involving humans is inherently not fully deterministic---and this applies in general to any interpretability method. For example, consider saliency methods or feature visualizations, both commonly used interpretability tools: the output of the algorithmic step (a saliency map, or a feature visualization, or a group of most activating images, etc.) must still be interpreted by a domain expert to be informative. Furthermore, two people might have different interpretations of a given saliency map depending on their prior knowledge of the domain, but in the case of saliency maps (unlike our setting!) it’s not even obvious how to evaluate how different the two “interpretations' are.
>
> In contrast, our approach makes this interpretation process more concrete by having the analyst 1) make and state an explicit hypothesis, and 2) validate the hypothesis counterfactually. In particular, we verify that the designed feature transformation distinguishes two learning algorithms according to Definition 2.
>
> **“How [would the method] benefit users in the three applications:”** While our work and case studies focused on identifying distinguishing features as an analytic or diagnostic tool, we expect that resulting insights can be valuable for downstream intervention. For instance, one can modify data to remove spurious correlation identified by the distinguishing feature. In the “human face bias” case, we can filter the dataset to remove images with humans in the background so that the model does not rely on this feature. In general, we believe applying insights from our algorithmic comparison framework will be valuable, but we believe these to be beyond the scope of this paper.
>
> We also note that each of the case studies we look at has been the subject of much more in-depth investigations in prior work (we discuss these under “Connections to prior work” within each case study). Rather than study each one in depth, our goal here is to illustrate how our framework can easily detect features and bias (some of which were previously known) across a variety of settings in a unified way.

---

> > ### Comment · Reviewer_1DZF · 2022-11-29
> > **my reply**
> >
> > I thank authors for their response. I have read through author response as well as other reviewers comments.
> > Concerns of me and reviewer SMZu regarding using datamodel for Stage I is resolved.
> > However, I choose to keep my score as weak reject because of the indeterminism of Stage II. For methods like saliency maps or feature visualization mentioned by authors, their algorithm outputs are deterministic, while the following human interpretation step is totally open. However, for the proposed method, the algorithm outputs of Stage II is inherently indeterministic due to the human-in-the-loop nature. This may give rise to problems such as "What if a human cannot find the pattern" (by SMZu) which is not appropriately addressed by authors.

---

> > > ### Author Response · Authors · 2022-12-01
> > > **Response to Reviewer 1DZF**
> > >
> > > Thank you for responding to our rebuttal. We would like to respond to a few points raised by the reviewer:
> > >
> > > 1. The possibility of "what if a human cannot find the pattern" in Stage II of our framework also applies to existing methods including saliency maps and feature visualization methods. The only difference is that for these methods, there is an “implicit Stage II” that does not appear in papers, but that humans are still expected to do if they want to derive utility from an interpretation.
> > > 2. In particular, even without Stage II entirely, we believe our contributions in this paper are still of general interest:
> > >    - Stage I of the framework is a fully deterministic procedure that produces an artifact (just as with saliency maps and feature visualization methods) that helps humans distinguish the two algorithms. In fact, even the first part of Stage II (constructing the distinguishing subpopulations) is deterministic (and thus one could view the output of our method as a set of distinguishing subpopulations—note that this is *exactly* the same output format as feature visualizations, which output a subpopulation of image corresponding to each neuron)
> > >    - Our formal definition of algorithm comparison (Definition 2) allows us to  *counterfactually verify* our results. This differentiates our work from other interpretability methods such as saliency maps, which, while deterministic, often fail basic counterfactual tests [1,2], lack model fidelity [3], and are hard to falsify [4].
> > >
> > > ---
> > >
> > > [1] Adebayo, Julius, et al. "Sanity checks for saliency maps." *Advances in neural information processing systems* 31 (2018).
> > >
> > > [2] Hooker, Sara, et al. "A benchmark for interpretability methods in deep neural networks." *Advances in neural information processing systems* 32 (2019).
> > >
> > > [3] Shah, Harshay, Prateek Jain, and Praneeth Netrapalli. "Do Input Gradients Highlight Discriminative Features?." *Advances in Neural Information Processing Systems* 34 (2021): 2046-2059.
> > >
> > > [4] Leavitt, Matthew L., and Ari Morcos. "Towards falsifiable interpretability research." *arXiv preprint arXiv:2010.12016* (2020).

---

> ### Author Response · Authors · 2022-11-18
> **final questions/concerns?**
>
> Thank you for taking the time to review our work. We hope that our rebuttal has addressed your questions and concerns. Please let us know if you have any unresolved concerns or additional questions about the paper or our rebuttal!

---

### Official Review · Reviewer_UoE7 · 2022-10-25

**Confidence:** 2
**Correctness:** 3
**Technical Novelty And Significance:** 3
**Empirical Novelty And Significance:** 3
**Recommendation:** 8

**Clarity, Quality, Novelty And Reproducibility:**


The paper is clear. It is organized well, and the techniques were
presented clearly.  There was a good discussion of related work.

**Strength And Weaknesses:**

Strengths:

-- The contribution can be readily seen to be useful (diagnostics,
 could be important for troubleshooting).

-- Mostly clear and easy to read paper.

Weaknesses (More details after the list):

-- Computational cost or complexity of the approach is not clear (eg
   each of the 3 steps in stage 1), and may be too expensive.

-- It would be good to have further experiments with regards to
 understanding the properties of the proposed technique:
 limits/weaknesses, strengths, say in discrimination power between two
 algorithms, etc.


* Regarding computational cost: you say that the expectation of the
model output on instances is taken with respect to random choises
within algorithm when training as well as instance (Definition 2).
But it's not clear how many times you had to sample and retrain, and
how that impacts the variance (stability) over the results you
observed. I looked at the appendix, and that didn't help much.


* With regards to how discriminating of your technique, among two
algorithms: If the two learning algorithms are very similar, does the
method find a useful difference? can one set up controlled experiments
changing the degree of similarity? For instance, the accuracies of the
two models can be nearly identical and relatively high, eg both at say
near 95%, and yet the two models could behave very differently.

How many dimensions of difference, via PCA, does the approach find?
(as a function of problem and algorithm characteristics)

-- Any thoughts on how your approach might help one diagnose a model's behavior (a single
 model) and remedy certain undesired model behavior?  (biases, etc).

typo:

Appendix B.   "small large rate"  --> small learning rate



**Summary Of The Paper:**


The authors present a way of comparing two learning algorithms by
figuring out which feature transformations/alterations, of test
instances, can alter the output (probability) of model trained by one
method (on average) while not changing the output of another (as
much).  This approach can shed light on how two different algorithms
(such as whether or how to use pretraining, or data-augmentation, ...) can
differ in specific learning problems, beyond just looking at model accuracy
(summary performance numbers) and beyond just looking at the model
behavior (outputs) on a specific (test) instances, but getting insights into how features
are being used differently (the patterns of emphasis on various features) by the two algorithms.

**Summary Of The Review:**


The approach is interesting in getting insights into how/why a model/algorithm behaves a certain way,
in part by comparing it to another model/algorithm! (and going deeper than just looking at final output
on specific instances).  I raised a few issues (see strengths/weaknesses), but overall I am positive.

---

> ### Author Response · Authors · 2022-11-15
> **Response to Reviewer UoE7 (Part 1)**
>
> We thank the reviewer for their detailed feedback and suggestions. We address each of the raised weaknesses below:
>
> **“Computational cost is not clear…and may be too expensive.”**
>
> The bulk of the computation of our method comes from the first step of estimating datamodels, which scales with the number of models trained and the time to train a single model. In our experiments, we trained up to 100k models for a given learning algorithm.
>
> To understand how our results change with the number of models used to estimate datamodels, we added the following analysis in Appendix F.7 *(new)*: we show that one can recover similar results using 10x fewer models than what we use in the paper currently, which suggests the method can be made significantly more efficient.
>
> Even scaled down this is still a heavy computational cost, but we believe that our framework is still useful for at least two reasons:
> - One, it allows us to discover fine-grained biases in an automated way without metadata or prior hypotheses. So while it is expensive, we think this is a worthwhile investment of computational resources, as it becomes increasingly important to understand and diagnose the biases of machine learning models.
> - Most of the computation involved is extremely parallelizable, which can significantly reduce the time depending on resources. For reference, with the latest optimizations (such as using the FFCV dataloader) estimating the CIFAR-10  datamodels in the paper takes 3 days on a single AWS machine (~\$800), so there is significant room for scaling up even on a modest budget.
>
> In this paper, we do not target efficiency explicitly as our goal is to explore the approach itself (comparing algorithms in terms of how they use the training data) and we want to avoid conflating statistical error (from insufficient models) with conceptual error (from the approach not capturing something). Lastly, because our method uses datamodels as a “black-box,” any improvements to efficiency of datamodel estimation in would automatically translate to faster algorithmic comparison.
>
> **(Rephrased) Understanding properties of our method: stability to randomness in training**
>
> As the reviewer observed, our goal is to model the model output function, which maps a given subset of the train set to the expected model output when trained on that subset. In practice, when estimating datamodels, we only train a single model for each sampled training subset (this is akin to observing only a single (noisy) output for each x in a linear regression problem). The averaging over randomness is done implicitly during the datamodel estimation across models trained on different subsets. For scalability with the number of models/subsets, see our answer to the previous point.
>
> **(Rephrased)  Understanding properties of our method: controlled setting**
>
> - *If model A and model B are similar, does it find a useful difference?*
> - *Can one setup a controlled study to vary level of similarity (e.g. both high accuracy + correct predictions but for different reasons)*
>
> We thank the reviewer for this suggestion. Given a set of learning algorithms whose induced models vary in their similarity, we expect our method to highlight differences that correlate with how different the two learning algorithms are.
>
> To evaluate whether this is the case, we added a controlled experiment in Appendix F.5 *(new)* that extends our case study on pretraining (Section 3.2) in two ways. First, we consider four algorithms that differ in one or more axes and control for (a) choice of pre-training dataset and (b) usage of pre-training. Second, we restrict our analysis to the majority group of the Waterbirds dataset (i.e., landbirds on land) to control for the test accuracies of models trained with different algorithms. We find that datamodels corresponding to ResNet18 and ResNet50 models pre-trained on ImageNet are more similar to each other than models trained from scratch or models pre-trained on synthetically generated data. Our findings demonstrate that our framework identifies training directions that (a) impact similar algorithms similarly and (b) distinguish dissimilar algorithms even when they result in models with similar (and relatively high) test accuracies.
>
> The effect of distinguishing feature transformations also seem to “interpolate” between two learning algorithms. For instance, in the data augmentation case study, the spider web bias, which is amplified via standard data augmentation, degrades smoothly as we lower the amount of cropping (more specifically, the minimum allowable random cropping ratio) used during training (see Appendix E.1). Although we did not explicitly compute datamodels for each of the learning algorithms corresponding to the different cropping ratios, we would expect that the explained variance of the “spider web” direction for a given learning algorithm would smoothly interpolate with its degree of random cropping.

---

> > ### Comment · Reviewer_UoE7 · 2022-11-18
> > **Novel promising work**
> >
> >
> > I have read all the authors' responses and the current reviews.   The authors have satisfactorily answered my concerns, and I believe most other issues raised by other reviews.  Computational cost may still remain a potential drawback but I think the work is sufficiently novel and
> > useful.  I'll raise my rating.

---

> ### Author Response · Authors · 2022-11-15
> **Response to Reviewer UoE7 (Part 2)**
>
> **“How many dimensions of difference, via PCA, does the approach find?”**
>
> Purely at a quantitative level, our analysis of PCA directions of residual datamodels (see Figures 3,5,7,19) shows that our method identifies many distinguishing training directions (in the sense that each direction has high explained variance for one learning algorithm but not the other).  However, only a fraction of these were readily interpretable in our “Stage 2” analysis. (That said, there were a number of other candidate subpopulations we identified but did not further analyze. See Appendix F.3 for examples.)
>
> We can interpret this observation---not every PCA direction surfacing a “meaningful” subpopulation---in two ways:
> - This is due to estimation error---as with datamodels, we are essentially trying to estimate vectors in high dimensions. Even in the ideal world where our data came from a generative model consisting of different subpopulations, estimation error would cause different subpopulations to be entangled in a given PCA direction, making it hard to interpret the direction.
> - There *is* a common feature specific to the direction, but we are just unable to identify it visually; a priori, there is no reason why models would only rely on features that are readily human interpretable.
>
> Understanding the above potential explanations better would be valuable for better understanding the limits of our algorithm.
>
> **“Any thoughts on how your approach might help one diagnose a [single] model’s behavior…and remedy undesired model behavior”**
>
> Our case studies demonstrate that our framework can semi-automatically identify specific spurious correlations without requiring any metadata. One straightforward way to apply these findings to improve the model is to use the identified feature transformations as data augmentation at train-time to prevent the models from relying on the spurious correlation. Alternatively, we can modify the training data to reduce undesired bias: in the example of the “human face bias”, we can filter the train set to remove images with human backgrounds (say by using a pre-trained human face classifier) to reduce the reliance on this spurious signal.
>
> We note that while our analysis is done on a collection/ensemble of models, we expect the diagnosis to apply to a single model as well, based on concentration (e.g., the error bars in counterfactual results of Figure 4 show that individual models’ behaviors are sufficiently distinct between the two learning algorithms with high probability).

---

### Official Review · Reviewer_SMZu · 2022-10-27

**Confidence:** 4
**Correctness:** 3
**Technical Novelty And Significance:** 2
**Empirical Novelty And Significance:** 2
**Recommendation:** 3

**Clarity, Quality, Novelty And Reproducibility:**

Clarity: The paper is overall clear, but probably misses important details, e.g., how is the datamodel computed, how scalable this method is, how sensitive is the framework to sample size.

Quality: The paper provides 3 interesting case study with concrete example which is good. On the flip side, it is unclear how generalizable the framework is and whether it would fail in other cases. The main idea seems to be an engineering idea more than a scientific idea.

Novelty: The idea is straightforward based on the idea of datamodel from the literature.

**Strength And Weaknesses:**

Strength: The paper provides several applications of this method, which seems interesting.

Weakness: It is unclear how exactly is the datamodel computed; It is unclear computation-wise whether this framework is practical in general settings; Regardless of computation front, it is unclear whether the framework can work in general settings, e.g., what if human can not find the patterns, does it also dependent on sample size?

**Summary Of The Paper:**

The paper proposes a human-in-the-loop framework that first tries to find most distinguishable samples for each algorithm, and then let human summarize observable patterns from those samples.

**Summary Of The Review:**

In summary, the paper seems to be an incremental work built on top of the datamodel idea. However, some details are missing in the paper and the paper is an interesting engineering idea rather than a scientific idea. Therefore, I do not think the paper meets the acceptance bar.

---

> ### Author Response · Authors · 2022-11-15
> **Response to Reviewer SMZu**
>
> We thank the reviewer for their comments and questions about our work. We address each of the reviewer’s raised points individually below.
>
> **“probably misses important details, e.g., …”**
> - **“how is the datamodel computed...”** We actually provide a detailed procedure on how to estimate the datamodels in Appendix C.3, so we are a bit confused at the reviewer’s comment here. Still, we have added end-to-end pseudocode for our method in Appendix C.3 and would be happy to add any details that the reviewer thinks are missing regarding the datamodel computation or any other part of the method.
> - **“How scalable this method is, how sensitive is the framework to sample size.”** We assume here that the reviewer is referring to the number of models that one needs to train in order to see the trends identified in the paper. To test this, in Appendix F.7 *(new)*, we show that one can recover similar results using 10x fewer models than what we use in the paper currently, which suggests the method can be made significantly more efficient if needed. In this paper we do not target efficiency explicitly, since our goal is to explore the approach itself (comparing algorithms in terms of how they use the training data) and we want to avoid conflating statistical error (from insufficient models) with conceptual error (from the approach not capturing something).
>
> In general, we agree that scalability is an important avenue for future work to explore—still, we think that our contributions in this work (a precise and quantitative definition of the comparison problem and a new approach for solving it) are significant on their own.
>
> **“What if a human cannot find the pattern:”** While our analysis (naturally) involved visual analysis, this can be replaced with the domain specific analysis. For example, if we’re analyzing a model for molecular property prediction, an expert can inspect or further analyze the subpopulations extracted from the stage I of our method to infer a distinguishing feature. Furthermore, we show in Appendix F.4 that this analysis step can be more automated through the use of an external model, such as CLIP.
>
> **”unclear how generalizable this is:”** Although we focused on comparing vision models in this work, our framework is agnostic to the modality or model classes being compared, and can be applied to any supervised learning setup. This is because our algorithmic stage is based on the datamodeling framework, which abstracts away the details of a model and analyzes how training data maps to predictions. We chose to focus on comparing image classifiers due to the large body of existing work that studies the different inductive biases of these models, allowing us to place our findings into context with existing literature. Image classification is also the standard setting in which prior work studies model comparisons (and is also a standard test bed for developing new modern machine learning techniques.)
>
> An interesting avenue for future work (and one that is uniquely allowed by our architecture-agnostic methodology) would be to study how design choices affect different “classical” model classes such as random forests or decision trees.
>
> **“incremental work on top of datamodels.”** We do not think that our work follows directly in any obvious way from the datamodels work. While our method critically leverages datamodel representations, the manner in which we apply them, the problem setup and goals, and the results we derive are all new and unlike anything we were able to find in the datamodels paper itself.
>
> **“more of an engineering idea than a scientific idea:”** The main contributions of our work are (a) coming up with the right framework to think about the problem of learning algorithm comparison, including formalizing the goal (i.e., finding distinguishing feature transformations) and (b) designing an algorithm to tackle this goal (i.e., residual datamodels, applying PCA). Both of these contributions are conceptual ones and require little to no engineering, but rather careful thinking about the problem and the right tools to apply. We did not know how to address this concern in our revision, but are happy to clarify or modify anything that will more clearly highlight the nature of our contribution.

---

> ### Author Response · Authors · 2022-11-18
> **final questions/concerns?**
>
> Thank you for taking the time to review our work. We hope that our rebuttal has addressed your questions and concerns. Please let us know if you have any unresolved concerns or additional questions about the paper or our rebuttal!

---

### Official Review · Reviewer_QoXq · 2022-10-27

**Confidence:** 3
**Correctness:** 3
**Technical Novelty And Significance:** 3
**Empirical Novelty And Significance:** 3
**Recommendation:** 5

**Clarity, Quality, Novelty And Reproducibility:**

Overall, the presentation of the paper is clear, and it is easy to read. The reproducibility of the technique is a bit hard since it involves a human-in-the-loop stage.

**Strength And Weaknesses:**

Strengths:
- Interesting technique in exploring the characteristics of the algorithm and the relation to the datasets for algorithm comparison.
- The technique is applicable to multiple scenarios.

Weakness:
- The authors claimed that the technique is applicable universally across different learning algorithms and any domain. However, within the paper, the authors only discuss neural network models and limit the technique to computer vision problems. Even in the human-in-the-loop stage, the technique involves visual inspection, which makes the technique only applicable to computer vision problems.
- The authors do not introduce any baselines in the experiments. Therefore, it's hard to assess the contribution of the paper completely.
- The technique involves a human-in-the-loop stage for designing feature transformation. However, there is no clear explanation of how to perform the step.


------
==Post rebuttal==

Thanks the authors for the feedback.
I have read the authors' feedback and other reviewer comments. The feedback addressed some of my concerns. However, some other concerns remain.

I think one of the weaknesses of the paper is in the human-in-the-loop part, where it makes the method not easily applicable to other cases. The method requires the analyst to infer a feature shared by given a subpopulation of inputs. Depending on the application, this stage may be hard to perform.

The application discussed in the paper is also limited to computer vison task. However, the authors branded the paper as a 'unified framework' for comparing (any) learning algorithms. I think the authors should provide more example from other domains, like text/tabular data, or tone down the claim they made.

Therefore, my recommendation remain the same.

**Summary Of The Paper:**

The authors propose a technique for comparing two machine learning algorithms in terms of the most important features/data that influence the prediction of the model the most. The technique is based on the datamodel representation model (Ilyas et.al., 2022). Based on the distinguished features produced by the model, the technique utilizes human-in-the-loop distinguished feature transformation that represents the characteristics of the compared algorithms. The authors tested the proposed technique on three case studies: data augmentation, pretraining, and hyperparameter tuning.

**Summary Of The Review:**

It's an interesting paper overall. But I have some concerns about its applicability. Therefore, I recommend weak rejection.

---

> ### Author Response · Authors · 2022-11-15
> **Response to Reviewer QoXq**
>
> We thank the reviewer for their comments and suggestions. We address each of the raised weaknesses individually below.
>
> **“The authors claimed … limit the technique to computer vision problems.”** Although we focused on comparing vision models in this work, our framework is agnostic to the modality or model classes being compared, and can be applied to any supervised learning setup. This is because our algorithmic stage is based on the datamodeling framework, which abstracts away the details of a model and analyzes how training data maps to predictions. We chose to focus on comparing image classifiers due to the large body of existing work that studies the different inductive biases of these models, allowing us to place our findings into context with existing literature. Image classification is also the standard setting in which prior work studies model comparisons (and is also a standard test bed for developing new modern machine learning techniques.)
>
> An interesting avenue for future work (and one that is uniquely allowed by our architecture-agnostic methodology) would be to study how design choices affect different “classical” model classes such as random forests or decision trees.
>
> **“Even in the human-in-the-loop stage … only applicable to computer vision problems.”** While stage 2 of our analysis for the specific case studies (naturally) involves visual analysis, this can be replaced with the appropriate domain specific analysis; one just needs a way to go from the extracted subpopulations to an inferred feature and a counterfactually testable hypothesis. For example, if we’re analyzing a model for molecular property prediction, an expert can analyze the subpopulations of molecules extracted from the stage I of our method, possibly using additional metadata, to infer a distinguishing feature (e.g., a common structural pattern). As an example, we show in Appendix F.4 (new) that this analysis step in our case study can be automated through the use of an external model such as CLIP (for a different non-vision modality, a similar tool can replace this).
>
> **“The authors do not introduce any baselines…”** To our knowledge, our paper is the first to provide a  formal definition of the algorithm comparison problem, which makes it difficult to directly apply a method from another paper as a baseline; for example, many of these methods output a numerical score, rather than a distinguishing feature. Moreover, it is often unclear how we can even reuse the ideas from other papers in a baseline—for example, we cannot (directly) use neural representations for model comparisons, as even representations of a fixed learning algorithm are not aligned. That is, two models trained with the exact same learning algorithm will have incomparable representations. This is in stark contrast to datamodel representations, which are naturally aligned—even for two different learning algorithms, the first coordinate of a datamodel representation has a consistent interpretation: the impact of the first training example on the model’s prediction.
>
>
> Nevertheless, we have added two baselines in Appendix F.6 (new):
> - First, we show that our method finds similar distinguishing directions even when we restrict our analysis to the subset of the dataset that  models trained with the two algorithms, on average, agree on. This shows that a simple disagreement-based analysis does not serve the same purpose as our proposed method, since our method finds distinguishing features even in the absence of any disagreement between models.
> - Second, we carefully adapt penultimate-layer representations to our setting and find that (a) penultimate-layer “residual representations” have low effective dimensionality (and thus very few non-trivial PCA components) and (b) our method applied on top of these adapted penultimate-layter representations does not extract training directions that clearly distinguish the two algorithms in terms of explained variance (see the added Appendix for more details here).
>
> **“Human-in-the-loop stage… no clear explanation.”** We thank the reviewer for pointing out that this can be made clearer. In our revision, we have clarified the exact task that the human has to perform in Stage 2: given a subpopulation of inputs, the analyst must infer a feature shared by them. Beyond this high-level goal, however, we intentionally left the human-in-the-loop stage rather general, since—just as the reviewer points out —different modalities will require different actions at this stage (e.g., for image classification, visual inspection was enough, but for audio signals, it could be that some spectral analysis is useful).
>
> The exact manner in which the analysis is done can also vary depending on external data or domain knowledge. For instance, one can test if the extracted subpopulations align with features extracted from metadata, or use an external model such as CLIP for analysis (see Appendix F.4 for an example of this approach).

---

> > ### Comment · Reviewer_QoXq · 2022-12-04
> > **Response**
> >
> > Dear authors.
> > Thank you for your detailed feedback.
> >
> > The feedback addressed some of my concerns.
> > However, some other concerns related to human-in-the-loop process and the area limitation of the paper still remain. Particularly since the authors branded the paper as a 'unified framework' for comparing (any) learning algorithms.
> >
> > Thank you.

---

> ### Author Response · Authors · 2022-11-18
> **final questions/concerns?**
>
> Thank you for taking the time to review our work. We hope that our rebuttal has addressed your questions and concerns. Please let us know if you have any unresolved concerns or additional questions about the paper or our rebuttal!

---

### Author Response · Authors · 2022-11-15
**General response**

We thank the reviewers for all their helpful comments, which we think have greatly improved the paper and highlighted areas where we could have been clearer. We hope to have addressed all of the raised concerns in our latest revision. We respond to each reviewer individually below, and will use this comment to highlight the (significant) additional experimentation we have done in response to reviewer concerns.

**Streamlining human-in-the-loop analysis with CLIP (new Appendix F.4)**. Many reviewers were concerned about the fact that our method involves a human-in-the-loop, whether because they felt it restricted our method’s applicability (Reviewer 1DZF) or because it hurts reproducibility (Reviewer QoXq). We respond to these concerns individually below—in summary, any explainability method will inevitably require a human in the loop to actually interpret the method’s output (for example, a SHAP or GradCAM attribution necessitates a human interpreter to infer what the method is highlighting). In our case (unlike these other cases), however, we are able to quantify the end-to-end success of the method via Definition 2 (distinguishing feature transformation).

We also left the human-in-the-loop stage intentionally vague to allow for different approaches based on the modality being studied and the side information (e.g., metadata or domain expertise) available. However, for the case of standard vision datasets, a new analysis in *Appendix F.4* shows that we can streamline the human-in-the-loop step by using CLIP to suggest the common feature as a natural language caption.

**Controlled experiment varying similarity of learning algorithms (new Appendix F.5)**. We design a controlled experiment based on our case study on ImageNet pre-training in Section 3.2. Our experiment compares four learning algorithms that differ in one or more axes to control for (a) usage of pre-training, (b) choice of pre-training dataset and (c) test accuracies of trained models.

**Baselines (new Appendix F.6)**. Reviewers QoXq and 1DZF both asked about the performance of our method compared to other baseline approaches. We point out below that our setting and approach are both substantially different (and indeed, somewhat incomparable) to previous ones, and we also include an new Appendix F.6 that rules out two of the most natural approaches to finding distinguishing transformations:
- First, we rule out approaches based on comparing model predictions, by showing that our technique finds distinguishing subpopulations even when we restrict our analysis to the subset of the dataset that the two algorithms agree on.
- Second, we find that even if we carefully adapt penultimate-layer representations to our setting, we are unable to recover the biases found by our method.

**Studying the effect of sample size (new Appendix F.7)**. Finally, two reviewers mentioned the high computational cost of training tens of thousands of models. While we wanted to eliminate finite-sample error for scientific purposes, we find in Appendix F.7 that our analysis in the pretraining case study (Section 3.2) remains unchanged even when we use a factor of 10 fewer models—this suggests that algorithm comparison may be possible using far fewer models than what we used in the main body of this paper.

**Other changes to the paper**. In addition to our new experiments, we also edited the paper in a few places for clarity and completeness—most notably, we clarified the human-in-the-loop step and added references to works about model debugging and bias finding (Appendix G).

---

We discuss each of these additional experiments as well as their related points in our individual responses below—we look forward to discussing any of these points more with the reviewers!

---

### Decision · Program_Chairs · 2023-01-20

**Decision:**

Reject

**Justification For Why Not Higher Score:**

The paper requires a major revision and thereby another round of peer review.

**Justification For Why Not Lower Score:**

N/A

**Metareview: Summary, Strengths And Weaknesses:**

This paper proposes a "unified" framework for comparing machine learning algorithms based on the datamodel representation model (Ilyas et.al., 2022). The proposed framework utilizes human-in-the-loop distinguished feature transformation that represents the characteristics of the compared algorithms. The paper demonstrates the applicability of the proposed framework on computer vision tasks.

Overall, a majority of expert reviewers have concerns about the general applicability of the proposed framework. Being branded as a "unified" framework, the reviewers expect to see a broader range of applications apart from computer vision, especially with respect to a human-in-the-loop component. The human-in-the-loop, which is the crucial component of this framework, will likely vary across different application domains, but it remains unclear how one can generalize this framework to other application domains.

As a result, I cannot recommend this paper in its current form for publication at ICLR 2023.


**Summary Of Ac-Reviewer Meeting:**

N/A